# CLEC5A is a critical receptor in innate immunity against Listeria infection

Szu-Ting Chen[1,2,3], Fei-Ju Li[1,4], Tzy-yun Hsu[4], Shu-Mei Liang[4,5], Yi-Chen Yeh[6], Wen-Yu Liao[4], Teh-Ying Chou[1,6], Nien-Jun Chen[2,7], Michael Hsiao[4], Wen-Bin Yang[4] & Shie-Liang Hsieh[1,4,8,9]

The C-type lectin member 5A (CLEC5A) is a pattern recognition receptor for members of the *Flavivirus* family and has critical functions in response to dengue virus and Japanese encephalitis virus. Here we show that CLEC5A is involved in neutrophil extracellular trap formation and the production of reactive oxygen species and proinflammatory cytokines in response to *Listeria monocytogenes*. Inoculation of *Clec5a*[−/−] mice with *L. monocytogenes* causes rapid bacterial spreading, increased bacterial loads in the blood and liver, and severe liver necrosis. In these mice, IL-1β, IL-17A, and TNF expression is inhibited, CCL2 is induced, and large numbers of CD11b[+]Ly6C[hi]CCR2[hi]CX3CR1[low] inflammatory monocytes infiltrate the liver. By day 5 of infection, these mice also have fewer IL-17A[+] γδ T cells, severe liver necrosis and a higher chance of fatality. Thus, CLEC5A has a pivotal function in the activation of multiple aspects of innate immunity against bacterial invasion.

[1] Institute of Clinical Medicine, National Yang-Ming University, 155 Li-Nong Street, Section 2, Beitou, Taipei 112, Taiwan. [2] Genome Research Center, National Yang-Ming University, 155 Li-Nong Street, Section 2, Beitou, Taipei 112, Taiwan. [3] Department of Microbiology & Immunology, Taipei Medical University, 250 Wuxing Street, Shinyi, Taipei 11031, Taiwan. [4] Genomics Research Center, Academia Sinica, 128 Academia Road, Section 2, Nankang, Taipei 115, Taiwan. [5] Agricultural Biotechnology Research Center, Academia Sinica, 128 Academia Road, Section 2, Nankang, Taipei 115, Taiwan. [6] Department of Pathology and Laboratory Medicine, Taipei Veterans General Hospital, 201 Shipai Road, Section 2, Beitou, Taipei 112, Taiwan. [7] Department of Microbiology & Immunology, National Yang-Ming University, 155 Li-Nong Street, Section 2, Beitou, Taipei 112, Taiwan. [8] Department of Medical Research, Taipei Veterans General Hospital, 201 Shipai Road, Section 2, Beitou, Taipei 112, Taiwan. [9] Institute for Cancer Biology and Drug Discovery, Taipei Medical University, 250 Wuxing Street, Shinyi, Taipei 11031, Taiwan. Correspondence and requests for materials should be addressed to S.-L.H. (email: slhsieh@gate.sinica.edu.tw)

The myeloid C-type lectin 5A (CLEC5A), also known as myeloid DAP12-associated lectin-1[1], is a spleen tyrosine kinase (Syk)-coupled receptor abundantly expressed by monocytes, macrophages and neutrophils. Whereas CLEC7A detects beta-glucans[2] and has critical functions in host defense against *Candida albicans* invasion[3], CLEC5A has been shown to recognize mannose and fucose expressed on envelope proteins of *Flavivirus* family members, and has critical functions in the pathologies of dengue virus-induced lethal diseases[4] and Japanese encephalitis virus-induced neuroinflammation[5]. Activation of CLEC5A by dengue virus and Japanese encephalitis virus can result in secretion of proinflammatory cytokines such as TNF, IL-6, CXCL-8, and IP-10. Moreover, dengue virus activates the NLRP3 inflammasome via CLEC5A to induce the production of IL-1β and IL-18[6, 7]. We have previously shown that dengue virus activates CLEC5A to upregulate osteolytic activity and dysregulate bone homeostasis[8]. Blockade of CLEC5A by anti-CLEC5A monoclonal antibody (mAb) not only attenuates dengue virus-induced and Japanese encephalitis virus-induced lethality, but also suppresses dengue virus-mediated activation of the NLRP3 inflammasome and osteolytic activity[4, 8]. These findings suggest that CLEC5A has critical functions in flavivirus-induced inflammatory reactions. However, whether CLEC5A is involved in the pathogenesis of other microbial infections has not been investigated.

Via phagocytosis and production of reactive oxygen species (ROS), macrophages, neutrophils and CCR2+ inflammatory monocytes are the first line of host defense against bacterial invasion[9, 10], and depletion of these cells increases susceptibility to systemic Listeriosis[11]. *Listera monocytogenes* is an intracellular Gram-positive bacterium that can cause disease in immuno-compromised individuals. In animal models, *L. monocytogenes* can induce systemic sepsis and liver abscess[12]. In addition to phagocytosis and ROS production, neutrophils produced web-like structures known as neutrophil extracellular traps (NETs), which comprise decondensed chromatin, histones, and subsets of granules and cytoplasmic proteins to ensnare variety of microbes[13]. Moreover, activated macrophages secrete proinflammatory cytokines (such as IL-1β, TNF and IL-6) and limit growth and spreading of bacterial populations[12]. IL-17A-producing γδ T cells have been shown to have a critical protective function against *Listeria* infection[14]: however, the pattern recognition receptors (PRRs) responsible are unclear. MyD88-deficient mice are highly susceptible to *L. monocytogenes* infection, suggesting that Toll-like receptors (TLRs) may be involved in host recognition of *L. monocytogenes*. Although TLR2 can recognize components of Gram-positive bacteria, TLR2 does not contribute significantly to host survival after Listeria infection[12, 15]. Thus, the key PRR responsible for resistance to *L. monocytogenes* infection is unknown.

Here we show that CLEC5A is a promising candidate PRR in macrophage-mediated and neutrophil-mediated defense against *L. monocytogenes*. CLEC5A not only has important functions in the production of free radicals and cytokines, but also drives NET formation (NETosis). Compared to wild-type (WT) littermates, the expression of IL-1β, TNF, and IL-17A is downregulated in *Clec5a*−/− mice, an effect associated with rapid bacterial spreading and increased bacterial loads in liver and blood. Moreover, the livers of *Clec5a*−/− mice develop severe necrosis associated with fewer IL-17A+ γδ T cells and massive infiltration of CCR2+ inflammatory monocytes. Together, these data indicate a critical role for CLEC5A in *L. monocytogenes*-induced innate immunity.

## Results

**CLEC5A is involved in *L. monocytogenes*-induced IL-1β production.** Macrophages are principle effector cells in the clearance of bacterial infections by exerting several bactericidal actions[12] and coordinating innate immune responses[16]. Among the proinflammatory cytokines secreted from macrophages after bacterial infection, IL-1β is crucial for induction of a febrile response and differentiation of T_H17 and IL-17A-producing γδ T cells[17, 18]. It has been reported that bacteria promote IL-1β production via TLR-dependent NF-κB[19] activation and Syk-mediated signaling[20]. Thus, we asked whether CLEC5A is involved in bacteria-induced inflammasome activation. To address this question, *L. monocytogenes* were incubated with WT, *Clec5a*−/− and *Tlr2*−/− macrophages, followed by examination of Syk phosphorylation and NF-κB activation. We found that *L. monocytogenes* induced the phosphorylation of Syk (Fig. 1a) and p65 (Fig. 1b) in WT macrophages; these responses were significantly impaired in *Clec5a*−/− and *Tlr2*−/− macrophages, respectively (Fig. 1a, b). Compared to WT macrophages, *Clec5a*−/− and *Tlr2*−/− macrophages produced less IL-1β after *L. monocytogenes* stimulation (Fig. 1c). In addition, IL-1β mRNA expression was downregulated in *Clec5a*−/− and *Tlr2*−/− macrophages incubated with live or UV-inactivated *L. monocytogenes* (Fig. 1d). These data indicate that both TLR2 and CLEC5A are involved in IL-1β production upon challenge with *L. monocytogenes* and that is independent of intracellular replication of the bacteria. It is interesting to note that caspase I activity is only downregulated in *Clec5a*−/− macrophages (Fig. 1e, f), indicating that CLEC5A, but not TLR2, is responsible for *L. monocytogenes*-induced caspase I activation. As CLEC5A is a DAP12-associated receptor[1], we asked whether DAP12 is required for *L. monocytogenes*–induced IL-1β production. As shown in Supplementary Fig. 1a, b, IL-1β production was downregulated dramatically in *Dap12*−/− macrophages, suggesting that CLEC5A-mediated IL-1β production during *L. monocytogenes* infection is mediated via DAP12.

It has been demonstrated that *L. monocytogenes* is sensed by multiple inflammasomes (including NLRP3, NLRC4, and AIM2 inflammasomes) that collectively orchestrate the production of IL-1β[21–23]. Compared to WT macrophages, *nlrc4* transcript was downregulated in *Clec5a*−/− macrophages while *aim2* transcript was downregulated in both *Clec5a*−/− and *Tlr2*−/− macrophages following *L. monocytogenes* infection (Fig. 1g). In contrast, the expression of *Nlrp3* transcript was similar in WT, *Clec5a*−/−, and *Tlr2*−/− macrophages.

Together, these observations suggest that CLEC5A and TLR2 are involved in regulating the transcription of IL-1β and inflammasome components, and that the CLEC5A-DAP12 signaling cascade regulates canonical caspase I-dependent inflammasome activation, in response to *L. monocytogenes*.

**CLEC5A is critical in neutrophil-mediated defense mechanisms.** We went on to test whether CLEC5A is involved in neutrophil-mediated immune responses to *L. monocytogenes*, i.e., phagocytosis, cytotoxicity, and ROS production, using Ly6G^hi neutrophils, from *Clec5a*−/−, *Tlr2*−/−, and WT mice. While the phagocytic activity of *Clec5a*−/− Ly6G^hi neutrophils was similar to WT (Supplementary Fig. 2a), *Clec5a*−/− and *Tlr2*−/− neutrophils had impairments in both cytotoxic activity (Fig. 2a) and production of ROS (Fig. 2b, c; Supplementary Fig. 2b). ROS production is not only an essential bactericidal action, but also contributes to bacteria-induced NET formation[24, 25]. Therefore, we further investigated the role of CLEC5A in NET formation by incubating human neutrophils with *L. monocytogenes* in the presence of an antagonistic anti-CLEC5A mAb[4], an isotype control or an anti-TLR2 blocking mAb[26]. Examination and quantitation of NET formation revealed that *L. monocytogenes* induced NET formation 3 h post infection, and that this was almost completely abolished by blockade of CLEC5A (Fig. 2d, e).

In contrast, blockade of TLR2 had a much weaker inhibitory effect on NET formation (Fig. 2d, e). We further validated these observations using $Clec5a^{-/-}$ and $Tlr2^{-/-}$ neutrophils. NET formation was detected in murine Ly6G[hi] neutrophils and peaked at 90 min post-infection with *L. monocytogenes* (Supplementary Fig. 3). In contrast, NET formation was impaired in both $Clec5a^{-/-}$ (Fig. 2f) and $Dap12^{-/-}$ (Supplementary Fig. 1c) neutrophils; mild impairment of NET formation was noted in $Tlr2^{-/-}$ neutrophils under the same conditions (Fig. 2f). To further explore the mechanism of CLEC5A-mediated NET formation, we examined peptidylarginine 4 (PAD4)-mediated histone citrullination[27], which causes histone decondensation and enhanced NET formation[28]. We found that histone 3 citrullination was severely impaired in $Clec5a^{-/-}$ neutrophils, but not in $Tlr2^{-/-}$ or WT neutrophils (Supplementary Fig. 4). Thus, CLEC5A contributes significantly to NET formation via ROS production and histone citrullination during *L. monocytogenes* infection.

**Increased bacterial spreading in *Clec5a*-deficient mice.** Since NET formation is known to limit bacterial spreading in vivo, we asked whether CLEC5A deficiency impairs this aspect of host immunity. Mice were inoculated intravenously (i.v.) with live luminescent *L. monocytogenes* (Xen32) and livers were collected to determine the extent of NET formation 4 h later. In WT livers, fibrous DNA structures associated with citrullinated histone 3 (mean NET histone area: 23972 µm²) and the neutrophil-specific marker myeloperoxidase (MPO) were detected (Supplementary Fig. 5a, b); digestion of the fibrous DNA with DNase I did not significantly reduce citrullinated histone 3 (mean NET histone area: 20,153 µm²) (Supplementary Fig. 5a, b). While only mild impairment of NET formation was observed in $Tlr2^{-/-}$ livers (mean NET histone area: 16580 µm²), citrullinated histone 3 (mean NET histone area: 6748 µm²) and fibrous DNA were reduced significantly in $Clec5a^{-/-}$ livers (Supplementary Fig. 5a, b).

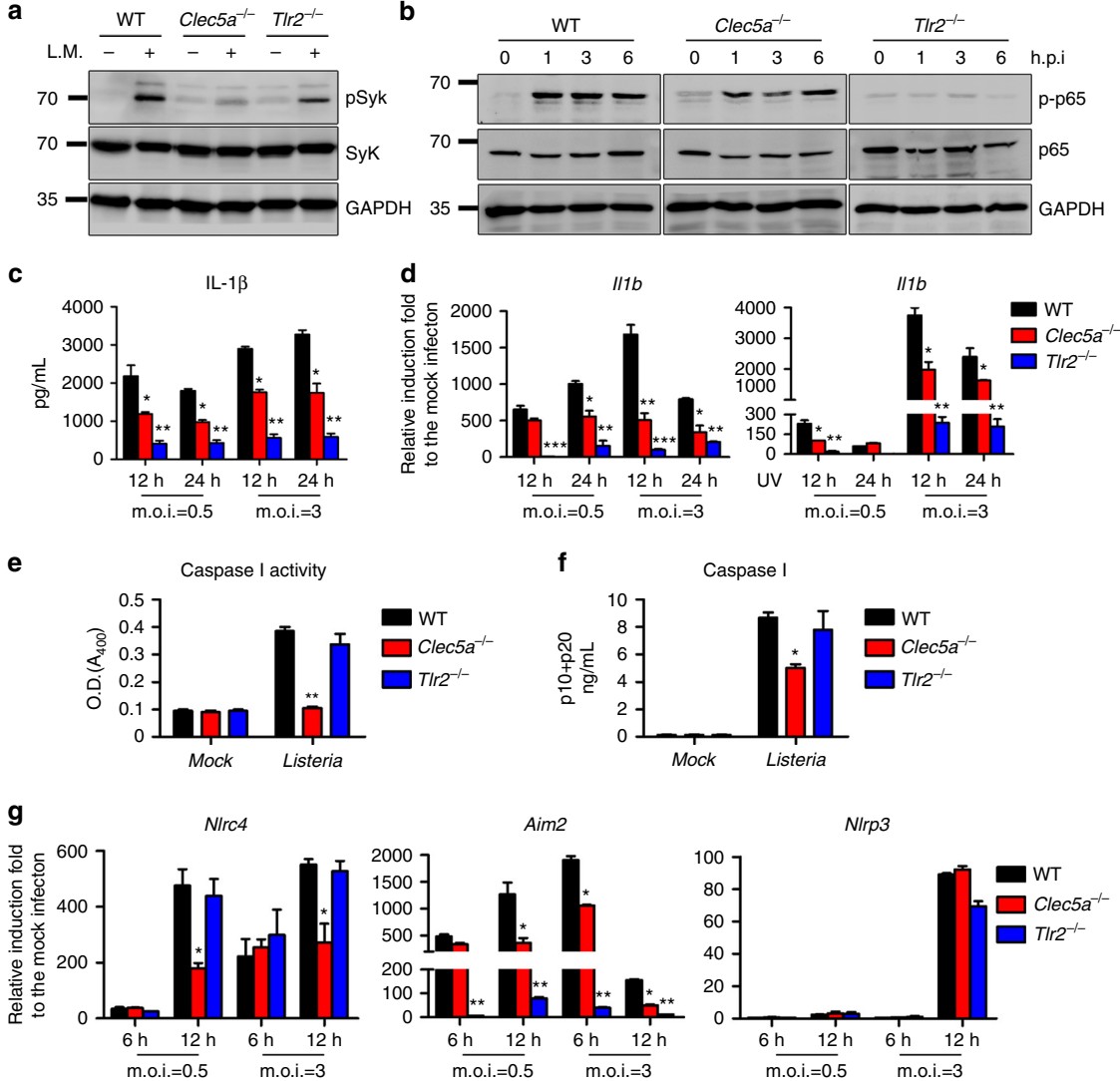

**Fig. 1** CLEC5A is involved in *L. monocytogenes*-induced IL-1β production. Mouse macrophages were incubated with live *L. monocytogenes* (MOI 10) and lysates were collected at 60 min post infection to detect **a** Phospho-Syk, while **b** phospho-NFκB p65 were detected at indicated time points. **c** Mouse macrophages were incubated with live *L. monocytogenes*, supernatants were collected and IL-1β secretion was measured by ELISA. **d** Mouse macrophages incubated with live (*left*) or UV-inactivated *L. monocytogenes* were collected 6 and 12 h post infection to detect *Il1b* transcripts by RT-qPCR. **e** Caspase-1 activity in mouse macrophages (6 h post Listeria infection) was presented as absorbance at 405 nm ($A_{405}$). **f** The amounts of caspase-1 p10 and p20 released from macrophages were determined by ELISA at 12 h post *L. monocytogenes* infection. **g** Mouse macrophages incubated with live *L. monocytogenes* were collected to detect *Nrlp3, Aim2*, and *Nlrc4* transcripts by RT-qPCR. All the data were collected and expressed as mean ± s.e.m. from at least three independent experiments. One-way ANOVA was performed. *$P < 0.05$, **$P < 0.01$; for treated versus control or knockout versus wild-type mice

Inactivation of *L. monocytogenes* in vivo is known to be mediated by neutrophils at 2–6 h after *i.v.* inoculation[29–32]. Here we investigated (using a minimally invasive imaging system) whether impaired NET formation in *Clec5a*−/− mice resulted in enhanced bacterial growth and spreading. In WT and *Tlr2*−/− mice, luminescence intensity in the liver peaked at 2 h post-infection with Xen32, before decreasing steadily (Fig. 3a, b). In contrast, luminescence intensity was still increasing at 4 h

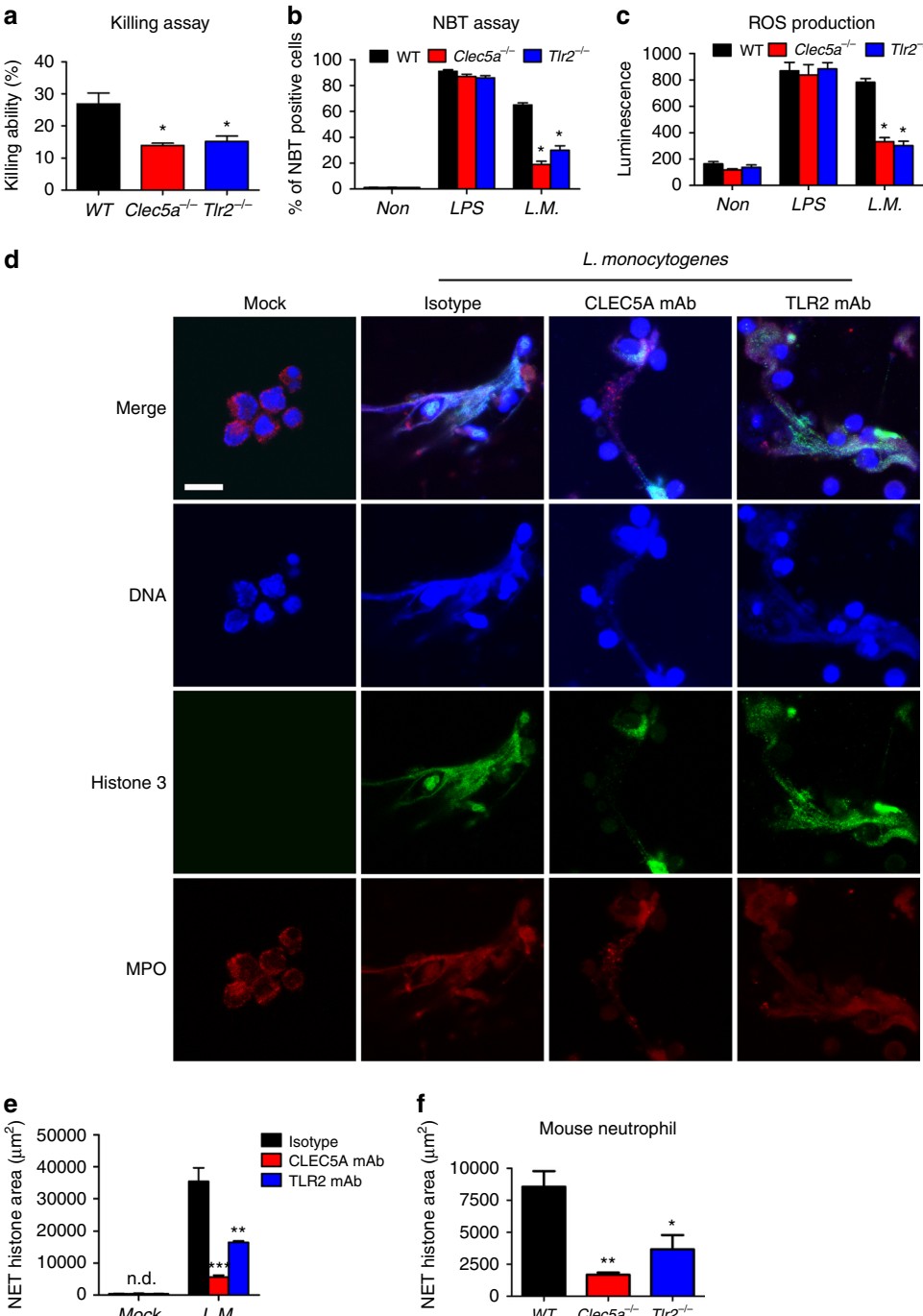

**Fig. 2** CLEC5A is critical for *L. monocytogenes*-induced NETosis and ROS production. **a** Mouse neutrophils were incubated with live *L. monocytogenes* (*L.M.*) (MOI 0.1) for 60 min to determine killing ability. **b** Neutrophils were co-incubated with *L. monocytogenes* (MOI 3) and NBT reagent for 60 min, followed by observation under a light microscopy to determine numbers of NBT-positive cells from the average of five individual fields in each group. **c** ROS production was measured by luminescence at 10-min intervals for 120 min, and the representative ROS production in each group is shown as luminescence intensity at 60 min post *L. monocytogenes* infection. **d** Human neutrophils were pretreated with anti-CLEC5A and anti-TLR2 mAbs (3 μg/ml), respectively, at room temperature for 30 min before incubation with *L. monocytogenes* (MOI 5). NET structures were analyzed 180 min post-infection by immunofluorescence staining using Hoechst 33342 (*blue*), anti-MPO (*red*) and anti-histone (*green*) mAbs. *Scale bar*, 30 μm. **e** The NET quantification is displayed as NET histone area (μm²) /per filed. **f** Mouse neutrophils were incubated with bacteria (MOI 10) for 90 min, and NET formation was quantified by determining histone area (μm²)/per filed. All the data were collected and expressed as mean ± s.e.m. from at least three independent experiments. One-way ANOVA was performed. *$P < 0.05$, **$P < 0.01$, ***$P < 0.001$ for treated versus control or knockout versus wild-type mice

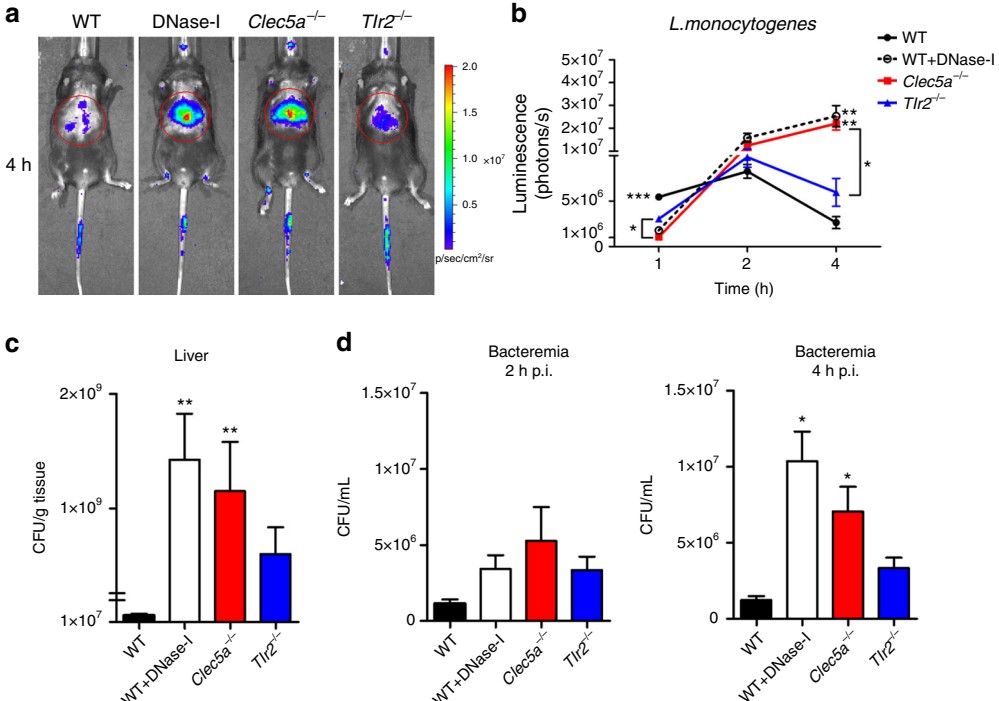

**Fig. 3** CLEC5A is critical for restriction of bacterial spreading in vivo. **a**, **b** Wild type (*WT*), WT/DNase-I (4 KU per mouse) and mutant mice were injected with the live luminescent *L. monocytogenes* (*Xen32*) ($3 \times 10^8$ CFUs per mouse) intravenously, and luminescence intensity in liver was determined at 1 h, 2 h, and 4 h post- infection (10 mice per group) by IVIS system (Xenogen). Bacterial loads in liver **c** and blood **d** were determined 4 h post-infection. The data were collected and expressed as mean ± s.e.m. from four independent experiments ($n = 20$). One-way ANOVA test performed. *$P < 0.05$, **$P < 0.01$, ***$P < 0.001$ for WT versus mutant mice, WT versus WT/DNase-I mice or indicated two groups

post-infection in the livers of WT/DNase I and *Clec5a*$^{-/-}$ mice (Fig. 3a, b) and this correlated with bacterial loads in the liver (Fig. 3c) and blood (Fig. 3d) at the 4 h time point. These observations suggest that CLEC5A-mediated NET formation contributes to clearance of *L. monocytogenes* infection.

**CLEC5A is critical for induction of IL-17A⁺ TCR γδ T cells**. We further investigated the impact of CLEC5A deficiency on systemic Listeriosis by administering a sublethal dose of *L. monocytogenes* to mice ($1 \times 10^5$ CFU per mouse, i.v.) and monitoring bacterial load in the liver. At day 1 post-infection, the highest bacterial load was observed in WT/DNase I mice (Fig. 4a). While bacterial load increased gradually in *Tlr2*$^{-/-}$ mice, it was significantly elevated in WT/DNase I, and *Clec5a*$^{-/-}$ mice at day 3 post-infection (Fig. 4a). By day 5, *Clec5a*$^{-/-}$ mice had the highest load in the liver and significant elevation was also seen in *Tlr2*$^{-/-}$ mice. This observation suggests that TLR2 contributes to limiting of *L. monocytogenes* proliferation in the later stages of infection.

Since *L. monocytogenes*-mediated activation of both CLEC5A and TLR2 induces phosphorylation of Syk and p65 (Fig. 1a, b), we investigated whether CLEC5A and TLR2 collaborate in host defense against systemic Listeriosis. *Clec5a*$^{-/-}$ mice were crossed with *Tlr2*$^{-/-}$ mice to generate *Clec5a*$^{-/-}$*Tlr2*$^{-/-}$ mice (Supplementary Fig. 6). We found that bacterial load in the liver was dramatically elevated in *Clec5a*$^{-/-}$*Tlr2*$^{-/-}$ mice at days 3 and 5 post-infection with *L. monocytogenes* (Fig. 4a). Similarly, bacterial load in peripheral blood was elevated in CLEC5A- and TLR2-deficient mice (*Clec5a*$^{-/-}$*Tlr2*$^{-/-}$ > *Clec5a*$^{-/-}$ > *Tlr2*$^{-/-}$) (Fig. 4b). These observations suggest that co-activation of CLEC5A- and TLR2-mediated pathways are required for optimal host immunity against systemic *L. monocytogenes* infection. To further test this

hypothesis, we examined the cytokine profiles in livers from *Clec5a*$^{-/-}$, *Tlr2*$^{-/-}$, and *Clec5a*$^{-/-}$*Tlr2*$^{-/-}$ mice post *L. monocytogenes* infection. We found that the expression of *tnf*, *Il-1β*, and *Il-17a* was downregulated in *Clec5a*$^{-/-}$ and *Tlr2*$^{-/-}$ mice (ANOVA, $P < 0.05$), while these cytokines were further downregulated in *Clec5a*$^{-/-}$*Tlr2*$^{-/-}$ mice (ANOVA, $P < 0.01$) (Fig. 4c). Interestingly, *ccl2* transcript was upregulated in *Clec5a*$^{-/-}$, *Tlr2*$^{-/-}$, and *Clec5a*$^{-/-}$*Tlr2*$^{-/-}$ mice (Fig. 4c).

IL-17A induction in the liver occurs very early post *L. monocytogenes* infection and this cytokine plays a critical role in host defense[14]. Here we determined the population of IL-17A-producing cells in liver at 5 days post *L. monocytogenes* infection. We found that in WT mice IL-17A was expressed by CD3⁺CD4⁻ TCRγδ⁺ T cells (1.65% of CD3⁺ cells; Fig. 4d, e), which have been shown to play a critical role in innate immunity against *L. monocytogenes* infection in liver[14]. In contrast, IL-17A-proudcing CD3⁺CD4⁻TCRγδ⁺ T cells were almost undetectable in *Clec5a*$^{-/-}$ mice (0.36% of CD3⁺ cells) and *Clec5a*$^{-/-}$*Tlr2*$^{-/-}$ mice (0.17% of CD3⁺ cells) (Fig. 4d, e). Even though the populations of IL-17A-producing γδ T cells were reduced in mice deficient in CLEC5A and TLR2, the total populations of CD3⁺ CD4⁺ T cells and CD3⁺CD4⁻ TCR γδ T cells were similar in all groups of mice tested (Supplementary Fig. 7). This suggests that CLEC5A and TLR2 are required specifically for the induction of IL-17A-producing γδ T cells, and not for the development of CD3⁺CD4⁻TCR γδ T cells, in the liver.

CLEC5A is not detectable in γδ T cells, even though it is highly expressed by myeloid cells[4] (Supplementary Fig. 8). Previous studies have shown that dengue virus can activate the NLRP3 inflammasome and induce the secretion of IL-1β[7], which is critical for the induction of an IL-17A-secreting CD3⁺CD4⁻ TCRγδ⁺ population[33]. Because IL-1β production is suppressed in *L. monocytogenes*-infected *Clec5a*$^{-/-}$ macrophages (Fig. 1c, d),

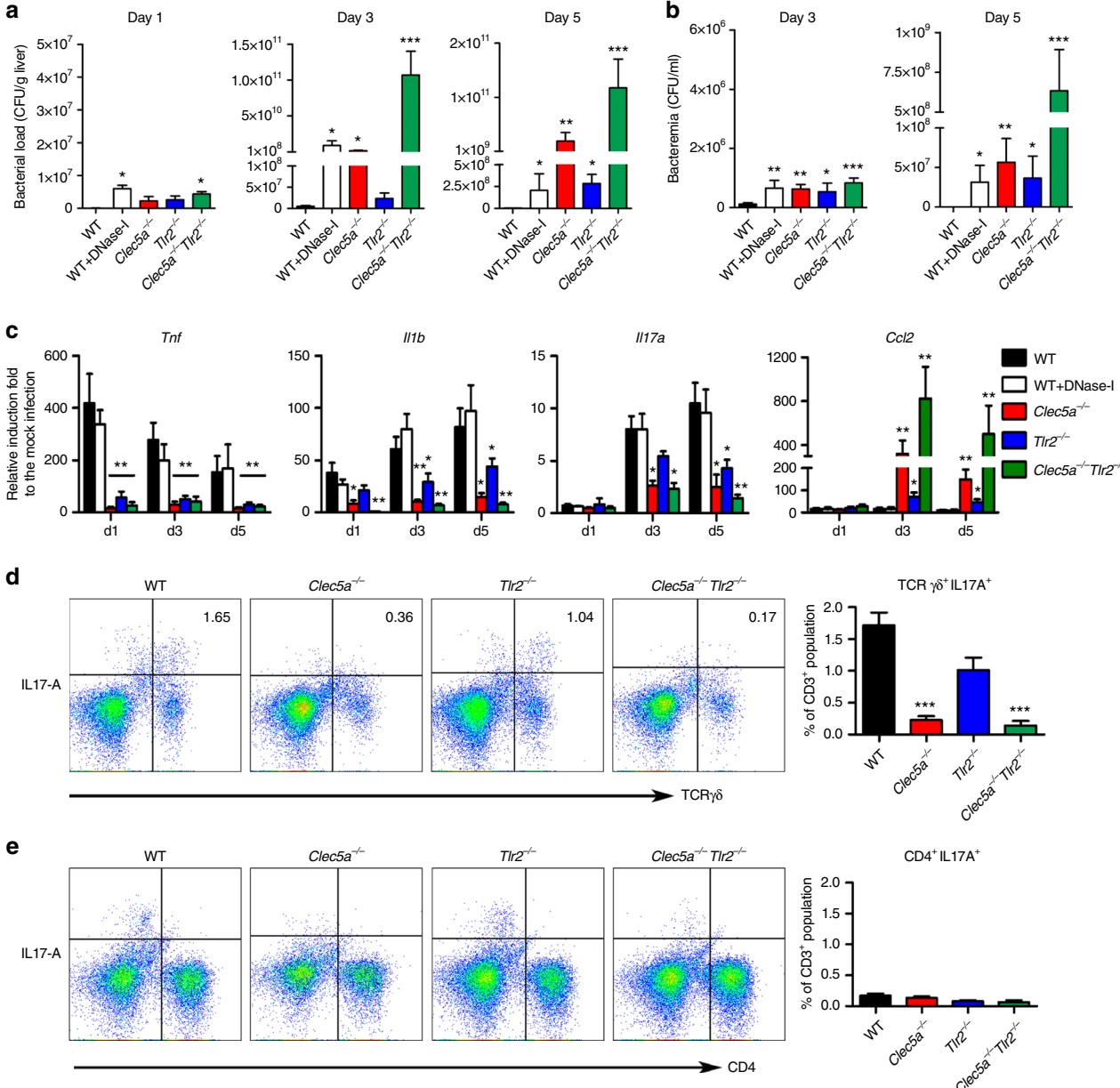

**Fig. 4** CLEC5A and TLR2 collaborate to eliminate *L. monocytogenes*. **a**, **b** All groups of mice were intravenously challenged with $1 \times 10^5$ CFUs of *L. monocytogenes* (10403S), and the bacterial loads in the livers **a** and blood **b** were determined at days 1, 3, and 5 post-infection (20 mice per group). **c** Livers were collected and lysed in TRIzol to extract total RNA for reverse-transcription into complementary DNA (cDNA) at the indicated time point post-infection. The expression levels of *Tnf*, *Il1b*, *Il17a* and *Ccl2* were determined by real-time PCR and normalized with *Gapdh*. The data were collected and expressed as mean ± s.e.m. from four independent experiments ($n = 20$). One-way ANOVA was performed. *$P < 0.05$, **$P < 0.01$, ***$P < 0.001$ for WT versus knockout mice, WT versus WT/DNase-I mice or indicated two groups. **d**, **e** *Left*: representative FACS plots of IL-17A-proudcing γδ T (CD3+CD4− TCR γδ+) and Th17 (CD3+CD4+TCR γδ−) cells under the CD3+-gated population in mouse liver at day 5 post *L. monocytogenes* infection. *Right*: The data were collected from three independent experiments ($n = 9$ for each group)

we concluded that the induction of IL-17A-secreting CD3+CD4− TCRγδ+ population during *Listeria* infection occurs via CLEC5A-dependent IL-1β production from myeloid cells, rather than direct activation of CD3+CD4−TCRγδ+ cells.

**CLEC5A deficiency results in liver inflammation and mortality**. Because CCL2 is critical for the recruitment of CCR2+ CD11b+Ly6c+ inflammatory monocytes[34], and is upregulated in *Clec5a*−/− mice after *L. monocytogenes* infection (Fig. 4c), we asked whether CLEC5A deficiency affected the recruitment of CCR2+CD11b+Ly6c+ cells to the liver. Compared to WT

littermates, extensive distribution of inflammatory foci with Ly6B.2+ cells were noted in the livers of knockout mice (*Clec5a* −/−*Tlr2*−/− > *Clec5a*−/− > *Tlr2*−/− mice) (Fig. 5a). Flow cytometry further demonstrated that the majority of the infiltrating cells were CD11b+Ly6c+ cells (monocytes), rather than CD11b+Ly6g+ cells (neutrophils) (Fig. 5b). In addition, the expression of CCR2 was increased in the livers of *Clec5a*−/− and *Clec5a*−/−*Tlr2*−/− mice (as determined by RT-PCR; Supplementary Fig. 9a), and multi-color flow cytometry revealed elevated numbers of CD11b+ Ly6c^hiCCR2^hiCX3CR1^low inflammatory monocytes among the infiltrating CD11b+Ly6c+ cells in these animals (WT: 18%,

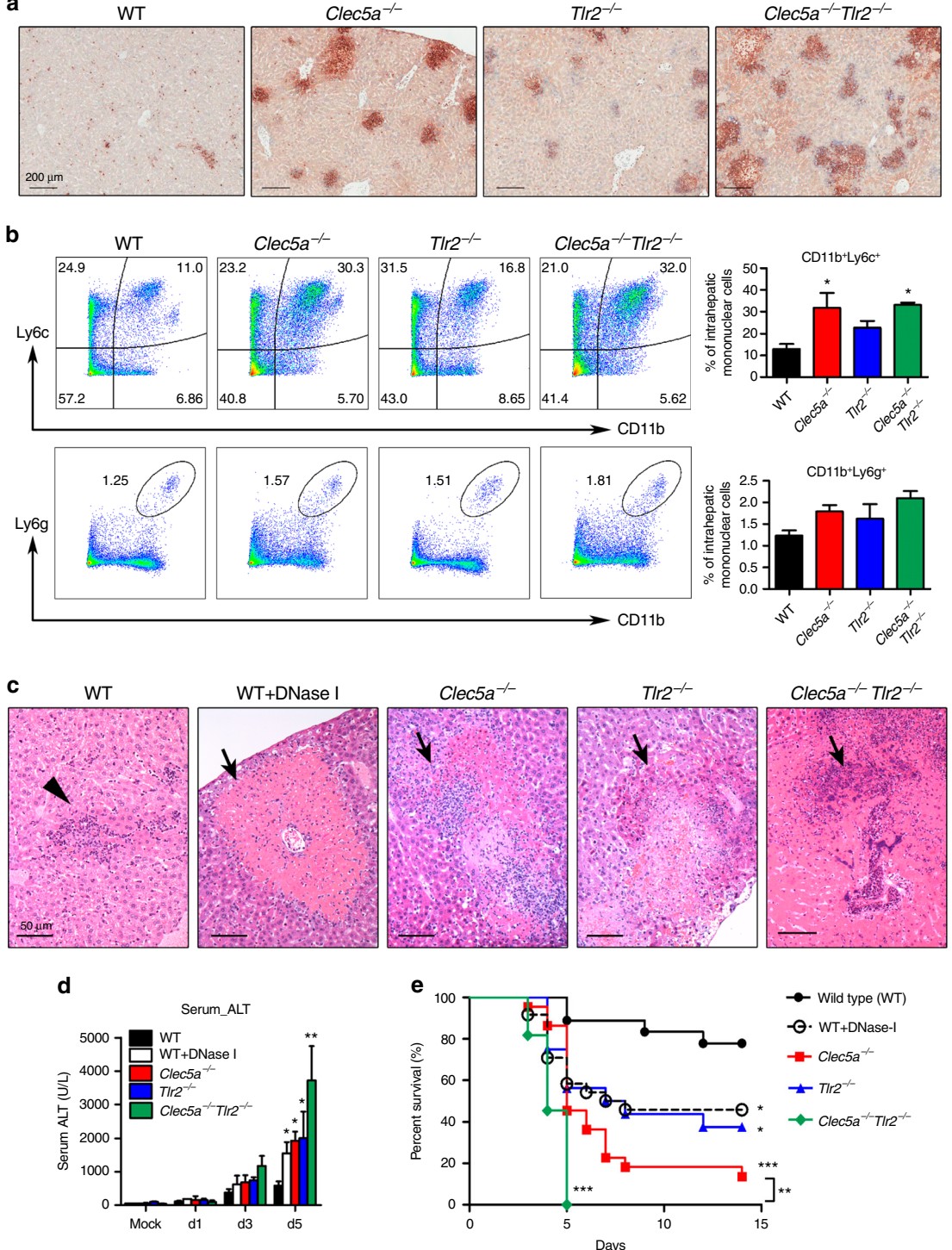

**Fig. 5** CLEC5A deficiency results in severe liver inflammation and high mortality. **a**, **b** All groups of mice were intravenously challenged with $1 \times 10^5$ CFUs of *L. monocytogenes* (10403S). **a** Livers were collected and fixed at day 5 post-infection, followed by immunohistochemical staining using anti-Ly6B.2 antibody to determine the presence of Ly6B.2+ cells. **b** *Left*: representative FACS plots of CD11b+Ly6c+ monocytes and CD11b+Ly6g+ neutrophils in mouse liver at day 5 post *L. monocytogenes* infection. *Right*: The data were collected from three independent experiments ($n = 9$ for each group). **c** H&E staining of liver section. *Arrowhead*: abscess. *Arrow*: region of necrosis (5 mice per group). **d** Alanine aminotransferase activity in serum ($n = 20$). **e** Survival rate was assessed daily for 15 days. The data were collected from four independent experiments, and the percentage of survived mice was displayed as Kaplan–Meier survival curves with log rank test ($n = 20$ for each group). The data were collected and expressed as mean ± s.e.m. from four independent experiments ($n = 20$). One-way ANOVA was performed. *$P < 0.05$, **$P < 0.01$, ***, $P < 0.001$ for WT versus knockout mice, WT versus WT/DNase-I mice or indicated two groups

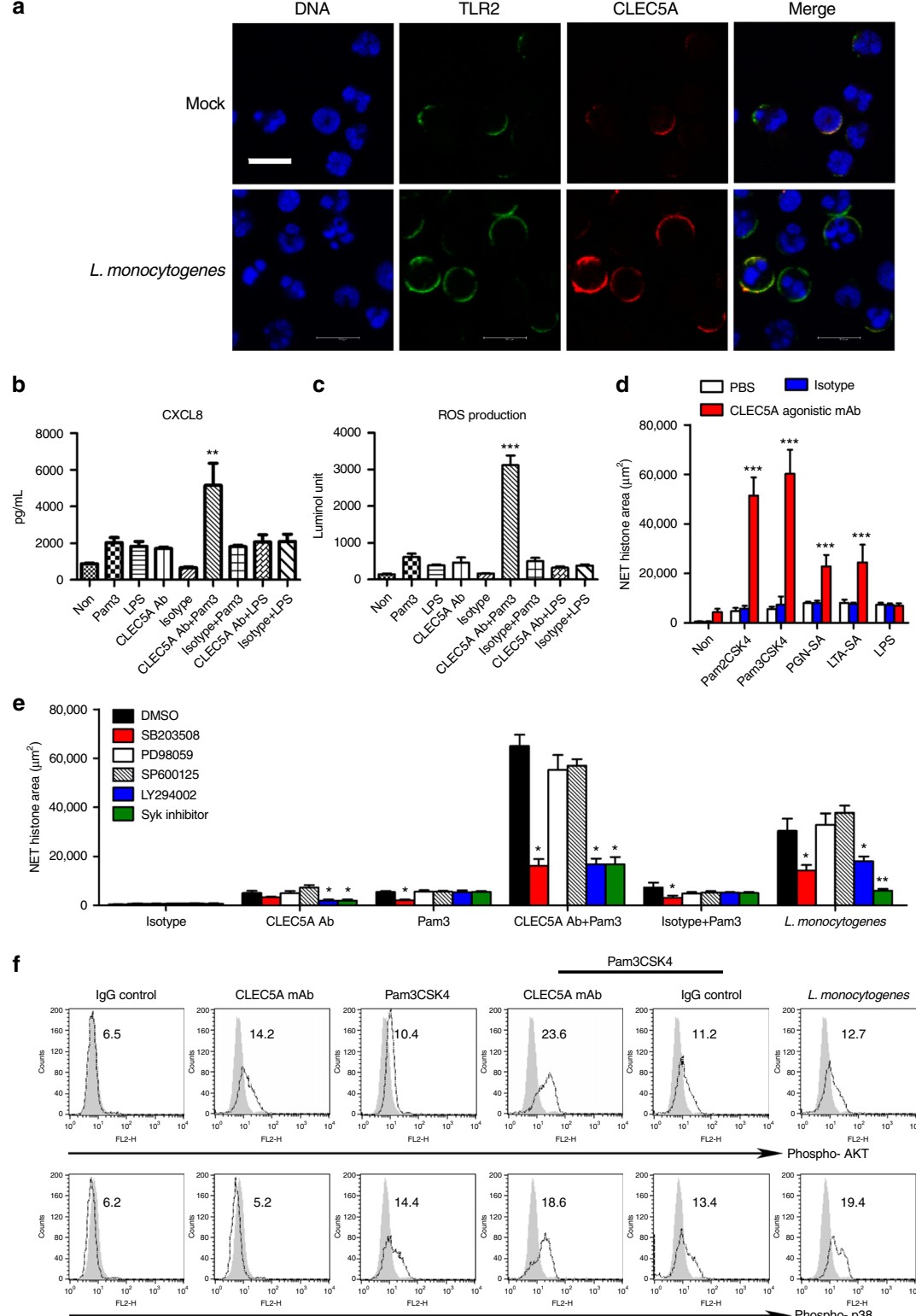

**Fig. 6** Coactivation of CLEC5A and TLR2 promotes NET formation via AKT and p38 MAPK. **a** Human neutrophils were incubated with *L. monocytogenes* for 60 min, and surface expression of CLEC5A and TLR2 was observed with a confocal microscope. *Scale bar*, 10 μm. **b**, **c** Human neutrophils were incubated with anti-CLEC5A agonistic mAb (2D8G9) and TLR ligands simultaneously for 2 h beforecollection, and **b** CXCL-8 production, **c** ROS production and **d** NET formation were determined. **e** Human neutrophils were pretreated with kinase inhibitors: p38 MAPK (10 μM, SB-203580), ERK (10 μM, PD-98059), JNK (10 μM, SP-600125), PI3K (5 μM, LY-294002), and Syk (10 μM, Syk inhibitor) for 30 min, followed by incubation with CLEC5A agonistic mAb, Pam3CSK4, or *L. monocytogenes*. NET was quantified by measuring histone and was displayed as NET histone area (μm²)/per filed. **f** Intracellular phosphorylated p38 MAPK and AKT kinases were detected by FACS at 30 min post stimulation. *Shaded* histograms: Isotype control; number: mean fluorescence intensity. For **b**–**e**, the data were collected and expressed as mean ± s.e.m. from four independent experiments. One-way ANOVA was performed. *$P < 0.05$, **$P < 0.01$, ***$P < 0.001$ for treated versus control groups (isotype or DMSO)

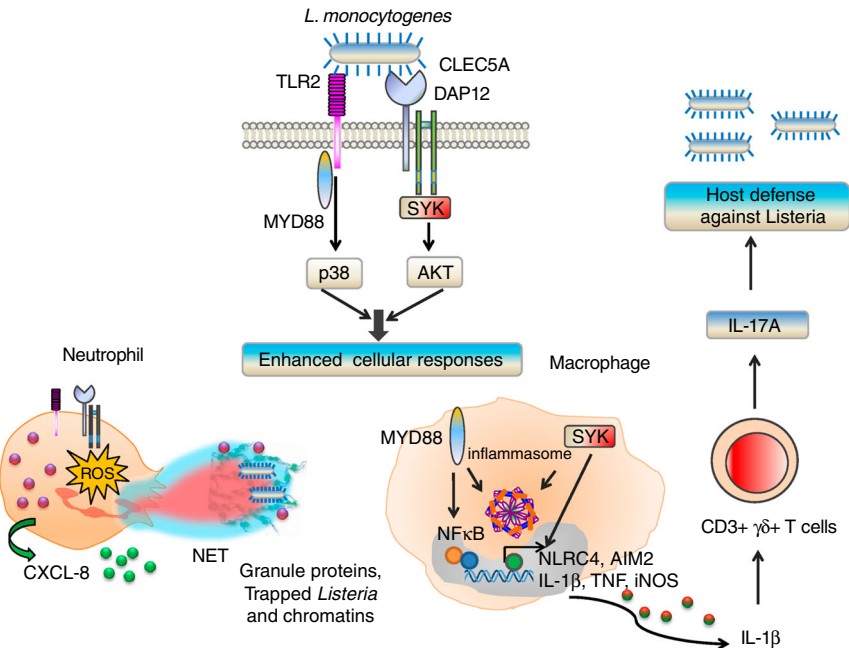

**Fig. 7** Co-activation of CLEC5A and TLR2 by *L. monocytogenes* enhances host immunity. Listeria activates CLEC5A and TLR2 simultaneously to activate both MyD88–p38 and Syk–AKT signaling pathways. Co-activation of CLEC5A and TLR2 in neutrophils enhances CXCL8 and ROS production, and NET formation, whereas co-activation of CLEC5A and TLR2 in macrophages induces inflammasome activation and production of IL-1β to stimulate the development of IL-17A prouducing TCR γδ T-cells

$Clec5a^{-/-}$: 28.7%, $Tlr2^{-/-}$: 23.2%, and $Clec5a^{-/-}Tlr2^{-/-}$: 33% of $CD11b^+CX3CR1^{low}$ cells) (Supplementary Fig. 9b). Thus, CLEC5A deficiency enhanced, rather than suppressed, the recruitment of inflammatory monocytes to the liver post *L. monocytogenes* infection.

Since $CCR2^+$ inflammatory monocytes mediate host defense against bacterial infection, we investigated whether this cell population was functionally impaired in $Clec5a^{-/-}$ and $Clec5a^{-/-}$ $Tlr2^{-/-}$ mice. Compared to WT macrophages, cytotoxic activity and ROS production were reduced dramatically in $Clec5a^{-/-}$ and $Clec5a^{-/-}Tlr2^{-/-}$ macrophages following incubation with *L. monocytogenes* in vitro (Supplementary Fig. 10). The production of IL-1β and TNF (Supplementary Figs. 11, 12), as well as the expression of inducible nitric oxidase synthase (iNOS), were also downregulated (Supplementary Fig. 13) post-infection in vivo. Furthermore, the extent of cell infiltration correlated with liver inflammation, necrosis (Fig. 5c and Supplementary Fig. 14), and elevation of serum aminotransferase (ALT) (Fig. 5d) in knockout mice ($Clec5a^{-/-}Tlr2^{-/-}$ > $Clec5a^{-/-}$ > $Tlr2^{-/-}$). These observations suggest that $Clec5a^{-/-}$ inflammatory monocytes are functionally impaired and unable to limit *L. monocytogenes* proliferation in vivo; CCL2 production in the liver is, therefore, elevated in order to attract more inflammatory monocytes and thereby compensate for the deficiency in bactericidal activity. The high mortality rate amongst the knockout mice post *Listeria* infection (Fig. 5e) was attributed to increased bacterial load and septic shock resulting from severe hepatocyte damage.

We went on to further examine the effect of attenuated NET formation by comparing the infiltrating cell populations in the livers of WT and WT/DNase I mice. We found that the majority of infiltrating cells were $CD11b^+Ly6c^+$ cells rather than $CD11b^+Ly6g^+$ cells (Supplementary Fig. 15a, b). Compared to WT mice, $CD11b^+Ly6c^{hi}CCR2^{hi}CX3CR1^{low}$ inflammatory monocyte numbers were elevated in WT/DNase I mice (WT: 20.4%, WT/DNase I: 31.5% of $CD11b^+CX3CR1^{low}$ cells; Supplementary Fig. 15c), suggesting that impaired NET formation also resulted in a compensatory recruitment of inflammatory monocytes to the

liver. Moreover, elevated serum ALT levels and an increased mortality rate were observed in WT/DNase I mice compared to WT following *L. monocytogenes* infection (Fig. 5d, e). Interestingly, adoptive transfer of WT neutrophils into $Clec5a^{-/-}$ mice partially reduced *L. monocytogenes* loads in blood and liver (Supplementary Fig. 16), further supporting the concept that NET formation contributes significantly to host defense against *L. monocytogenes* infection. Higher mortality rates in knockout mice ($Clec5a^{-/-}Tlr2^{-/-}$ and $Clec5a^{-/-}$) compared to WT/DNase I mice indicates that the mechanism of CLEC5A-mediated protection against *Listeria* infection involves both NET formation and cytotoxic effects of inflammatory monocytes and neutrophils.

**Dissection of CLEC5A/TLR2 co-activated signalling pathways.** Microorganisms express multiple pathogen-associated molecular patterns and it is likely that these activate multiple PRRs simultaneously to enhance host immunity. Consistent with this, we observed that both CLEC5A and TLR2 were upregulated and co-localized in neutrophils after incubation with *L. monocytogenes* (Fig. 6a). Moreover, co-activation of CLEC5A and TLR2 on neutrophils by agonistic anti-CLEC5A and TLR2 ligands not only enhanced chemokine production (Fig. 6b), but also further upregulated ROS production (Fig. 6c) and NET formation (Fig. 6d and Supplementary Fig. 17). Similar responses were observed in macrophages co-activated with agonistic anti-CLEC5A mAb and TLR2 ligand (Pam3CSK4) (Supplementary Fig. 18). It is interesting to note that NETosis was suppressed efficiently by inhibitors of p38 (SB203508), PI3K (LY294002), and Syk (Fig. 6e), where *L. monocytogenes* activates both AKT and p38 (Fig. 6f). Moreover, activation of CLEC5A induced phosphorylation of AKT (MFI = 14.2), but not p38 (Fig. 6f), while TLR2 ligands mainly induced p38 phosphorylation (MFI = 14.1) (Fig. 6f). In contrast, co-activation of both CLEC5A and TLR2 enhanced the phosphorylation of both AKT (MFI = 23.6) and p38 (MFI = 18.6) (Fig. 6f). These observations suggest that co-activation of CLEC5A and TLR2 by *L. monocytogenes* will

stimulate p38- and PI3K-Akt- pathways and thereby enhance host immunity against infection (Fig. 7).

## Discussion

In this study, we investigated the role of CLEC5A in the innate immune response to bacterial infection using *L. monocytogenes* (a Gram-positive, intracellular bacterium) as a model system. Our data reveal that CLEC5A plays a critical role not only in macrophage activation and cytokine production, but also in neutrophil-mediated cytotoxicity and NET formation. Elimination of NETs, via DNase I treatment, enhanced bacterial spreading in vivo (Figs. 3, 4a), suggesting that NET formation, which is suppressed in *Clec5a*[−/−] neutrophils, contributes significantly to host defense against *L. monocytogenes*. In addition, CLEC5A deficiency led to impaired production of TNF-α and IL-1β as well as reduced numbers of IL-17A-producing γδ T cells, which are critical for immunity to *L. monocytogenes*[14]. Moreover, *Clec5a*[−/−] mice were more sensitive than *Tlr2*[−/−] animals to infection with a sub-lethal dose of *L. monocytogenes*. All these observations provide compelling evidence that that CLEC5A is a critical PRR in *Listeria* infection.

NET formation is dependent on the proxduction of ROS, which enables the release of neutrophil elastase (NE) and MPO from azurophilic granules[35]. In addition, PAD4 is responsible for histone citrullination, which promotes decondensation of nuclear DNA[28]. ROS production was similar in *Clec5a*[−/−] and *Tlr2*[−/−] neutrophils, but PAD4 activity was impaired in *Clec5a*[−/−] neutrophils post *L. monocytogenes* infection. It has been shown that histone citrullination by PAD4 is facilitated by elevated ROS[36] and cytosolic calcium concentration[37, 38]. Since activation of CLEC5A induces both calcium mobilization[1] and ROS production (Fig. 6c), it is likely that the dominant role of CLEC5A, compared to TLR2, in promoting NET formation is associated with a PAD4-mediated mechanism.

Studies in humans and mice have shown that neutralization of IL-1 by IL-1R antagonist (IL-1RA; Kineret, which is used to treat rheumatoid arthritis) is associated with increased susceptibility to bacterial infections[39, 40]. This is consistent with the impaired immune response to *Listeria* in *Clec5a*[−/−] mice, where IL-1β production is suppressed. TLR2-deficient mice have reduced *Il-1β* expression compared to WT at 6 h post-infection with *Staphylococcus aureus*, but not at 24 h[41], suggesting that the role of TLR2 in IL-1β production is most significant in the early stages of infection. Although we observed less IL-1β production in *Tlr2*[−/−] than *Clec5a*[−/−] macrophages incubated with *L. monocytogenes* (Fig. 1c, d), *Tlr2*[−/−] mice were more resistant than *Clec5a*[−/−] mice to *Listeria* infection (Fig. 5e). Furthermore, the higher mortality rate and severe liver inflammation and necrosis in *Clec5a*[−/−] *Tlr2*[−/−] mice (compared to *Clec5a*[−/−] and *Tlr2*[−/−] animals) suggests that, while the DAP12-assoicated CLEC5A is a critical PRR for *L. monocytogenes* (Fig. 5e and Supplementary Fig. 19), co-activation of multiple CLEC5A- and TLR2-mediated signaling pathways is required to initiate a robust immune response to systemic *Listeria* infection.

Since IL-1β is critical for host defense against bacterial infections in general, we investigated whether CLEC5A is required for IL-1β production in response to other bacteria. We found that, compared to WT cells, *Clec5a*[−/−] macrophages produced less IL-1β after incubation with *S. aureus* (Gram-positive, extracellular bacterium) and *K. pneumoniae* (Gram-negative, extracellular bacterium), but not *S. typhimurium* (Gram-negative, intracellular bacterium) (Supplementary Fig. 20), suggesting that CLEC5A is an important PRR for *S. aureus* and *K. pneumoniae*. This is further supported by our observations of impaired NET formation (Supplementary Fig. 21) and ROS production (Supplementary Fig. 1b) in *Clec5a*[−/−] neutrophils after incubation

with *S. aureus* or *K. pneumonia*. Moreover, *Clec5a*[−/−] mice showed increased susceptibility to *S. aureus* infection, with rapid bacterial spreading to regional lymph nodes after subcutaneous inoculation (Supplementary Fig. 22).

Using a CLEC5A.Fc fusion in a microbial glycan microarray analysis[42], we found that CLEC5A also binds to other bacteria, including *Pseudomonas aeruginosa*, *Streptococcus pneumonia*, *Escherichia coli*, and *Shigella flexneri* (Supplementary Table 1). Since *L. monocytogenes* was not represented in the glycan array used here, cell wall components were extracted from *L. monocytogenes* and the CLEC5A.Fc fusion protein was shown to bind to this extract in a dose-dependent manner (Supplementary Fig. 23a). Moreover, fluorochrome-conjugated *L. monocytogenes*, *S. aureus*, and *K. pneumonia* (but not *S. typhimurium*) bound to 293T cells overexpressing CLEC5A and TLR2 (Supplementary Fig. 23b). These observations demonstrated the direct interactions of bacteria with CLEC5A and TLR2, further supporting the role of CLEC5A as a PRR in defense against a range of bacterial infections.

To determine whether CLEC5A binds to terminal sugars of *L. monocytogenes*, we extracted cell wall polysaccharides from various *L. monocytogenes* strains (see Supplementary Table 2). In all cases, we observed similar binding to CLEC5A.Fc by ELISA and similar ability to induce NET formation and cytokine production (data not shown). This observation suggests that terminal sugars (rhamnose, galactose, glucose) on wall teichoic acids do not play significant roles in the interaction of *L. monocytogenes* with CLEC5A and subsequent activation of the host immune response. In contrast, CLEC5A binding to *Listeria* cell wall extracts was inhibited by N-acetylglucosamine (GlcNAc) and N-acetylmuramic acid (MurNAc) disaccharides; this observation suggests that CLEC5A binds GlcNAc-MurNAc disaccharide backbone, rather than terminal sugar, of *L. monocytogenes*.

Here we have shown that CLEC5A is an important PRR in immune responses to bacteria, including *L. monocytogenes* and *S. aureus* via activation of macrophage-, neutrophil- and γδT cell-mediated effector functions. Previously, CLEC5A and other DAP12-associated receptors have been implicated in responses to mycobacterial infection *(Mycobacterium bovis)*[43]. Our observation that CLEC5A can promote differentiation of IL-17A-producing CD3[+]CD4[−] TCRγδ T-cells, which are necessary for granuloma formation during mycobacterial infection[44], might provide a mechanism for this activity.

## Methods

**Reagents**. Culture media/supplements and HBSS buffer were from Invitrogen GIBCO. Chemical reagents and chemical inhibitors were purchased from Sigma and Calbiocam, respectively. TLR ligands, including pam2csk4 (tlr-pm2s-1), pam3csk4 (tlr-pms), and LPS (tlr-eklps) were obtained from Invivogen. The monoclonal antibodies used in this study was listed as following: mouse IL-1β ELISA kit (DY 401), human TNF ELISA kit (DY210) and blocking antibody to human TLR2 (MAB2616) were purchased from R&D Systems, antibodies to histone H3 (citrulline R2 + R8 + R17; ab5103, 1:100), myeloperoxidase (MPO, sc-52707 or sc-34159, 1:100), phospho-syk (#2701, 1:1000), phosphor-p65 (#3033, 1:1000), histone 1–4 (Mab 3422, 1:200). Alexa Fluor® 488 Conjugated antibody against phosphor-p38 (#690, 1:100) and phosphor-AKT (Ser473, #2336, 1:100) were from Cell Signaling. CLEC5A antagonistic mAb has been characterized before[4].

**Bacterial strains**. Bacteria, including *L. monocytogenes* (10403S, Serotype 1/2a), *S. aureus* (25923), *S. typhimurium* (13311) were obtained from ATCC.
*K. pneumoniae* was from a clinical isolation (NTUH K2044, serotype K1).
*L. monocytogenes* Xen32 possessing a stable copy of the modified *Photorhabdus luminescens lux* operon at a single integration site on the bacterial chromosome was derived from the parental strain *L. monocytogenes* 10403S (Serotype 1/2a wild-type strain), was purchased from Caliper Life Science and used for the Xenogen IVIS optical imaging technology in this study. For serogroups of *L. monocytogenes* with different glycan substitutents on the cell wall teichoic acid structure, including EGDe (1/2a), EGDe (lack of rhamnoase), EGDe (lack of N-acetylglucosamine), EGDe (lack of rhamnoase, and N-acetylglucosamine; GlcNAc)[45], 4MT (serotype 4b, ATCC10527), XL7(lack of glucose on TA)[46], M44 (lack of galactose on TA)[47].

ATCC19111 (serotype 1), ATCC19112 (serotype 2), ATCC19113 (serotype 3), ATCC19114 (serotype 4a), ATCC19115 (serotype 4b) are from Dr Stowell Sean.

**Mice**. All experiments were conducted with mice housed under SPF conditions that were age matched (8–10 weeks old) and backcrossed for at least twelve generations onto the C57B/6 background. $Clec5a^{-/-}$ mice[5] were backcrossed for twelve generations into C57/B6 background; and $Tlr2^{-/-}$ mice (C57/B6) were from the Jackson Laboratory (Bar Harbor, ME). $Clec5a^{-/-}Tlr2^{-/-}$ mice (C57/B6) were generated by crossing $Clec5a^{-/-}$ and $Tlr2^{-/-}$ mice, and the F1 offspring were further interbred to generate F2 offspring (Supplementary Fig. 6). $Dap12^{-/-}$ mice (C57/B6) were from Dr NJ Chen (a gift from Dr Toshiyuki Takai, Department of Experimental Immunology, Institute of Development, Aging and Cancer, Tohoku University, Sendai, Japan). All gene-deficient mice and littermate control mice were bred in the animal facility within campus, and all animal experiments were in compliance with the Guide for the Care and Use of Laboratory Animals of the Taiwanese Council of Agriculture in a protocol approved by the Institutional Animal Care and Use Committee (IACUC) of Academia Sinica (IACUC #1510872).

**Neutrophil isolation**. For human neutrophil isolation, peripheral venous blood from the healthy donors was taken according to a protocol approved by the Academia Sinica Ethics Committee (AS-IRB-BM-14006). Blood samples were incubated with acid citrate dextrosefollowed by centrifugation to remove plasma and platelets before subjected to Ficoll-Paque (Amersham Biosciences) density-gradient centrifugation to isolate granulocytes. The erythrocytes were removed by two rounds of hypotonic lysis with RBC lysis buffer, while the viability and purity of neutrophils were determined by CD66 and 7-AAD double staining using FACS Caliber (Becton-Dickinson). Human neutrophil were obtained from healthy donors under a protocol (AS-IRB02-103202) approved by the IRB of the Clinical Center of the Department of Health, Taiwan. Written informed consent was obtained from all donors. Isolation of mouse neutrophils from the bone marrow was modified as described previously[48]. Briefly, bone marrow was flushed out of the tibia and the femur using RPMI medium supplemented with penicillin/streptomycin, and passed through a 70 µM cell strainer to obtain a single-cell solution. Cells were pelleted, and the remaining erythrocytes were lysed by RBC lysis buffer, followed by washing with Hank's balanced-salt solution (HBSS) without CaCl$_2$/MgCl$_2$. Bone marrow cells were further suspended in 45% (v/v) Percoll/HBSS and overlaid onto a discontinuous Percoll gradient comprising 52% (v/v), 62% (v/v) and 81% (v/v) Percoll in HBSS, followed by centrifugation at $1,000 \times g$ for 30 min at room temperature. Mouse Neutrophils were recovered from the interphase between layers of 62 and 81% Percoll, and washed with HBSS. The purity (>90%) of isolated neutrophils was confirmed by FACS analysis.

**Detection of NET structure by immunofluorescence staining**. Human and mouse neutrophils were suspended in RPMI with 10% of autologous serum and seeded on the coverslip, then allowed to settle for 2 h. Cells were either infected with bacteria or treated with TLR ligands followed by fixing cells with 4% paraformaldehyde at indicated time points. Fixed cells were subjected to permeabilization of cell membranes with 0.5% Triton X-100 in phosphate-buffer saline (PBS), then blocked with 3% bovine serum albumin in PBS for 1 h at room temperature, and incubated with primary antibodies against anti-histone and MPO at 4 °C overnight. Secondary antibodies conjugated with Alex488 or TRITC were applied to detect primary Ab, and Hoechst 33342 were used as a counter stain. NET images were observed under confocal microscope (Leica: TCS-SP5-MP-SMD), and images were analyzed by Leica LAS AF software. For the liver histology, livers were dissected and mounted in OCT embedding compound, and then frozen at -80 °C to generate 10-microm sections for immunofluorescence staining. Slides were fixed with cold acetone at 4 °C for 10 min, followed by the air dry for 30 min, and then incubated with Hoechst 33342 and primary antibodies: antibody to histone and MPO. Stained slides were detected by fluorochrome-conjugated secondary antibodies as mentioned above. Images were obtained by using Leica confocal microscope with white light laser system (TCS-SP5-MP-SMD). Image was analyzed by Leica LAS AF software.

**Quantification of NET formation**. NET formation was quantified by histone area. All images were obtained by using Leica confocal microscope (TCS-SP5-MP-SMD) and processed under Leica LAS AF software. Histone image of neutrophil were analyzed using MetaMorph® software. Briefly, NETs were visualized in at least five random fields (40 × magnification), and signal intensity of histone per fields was individually measured; and the pixels of each image were converted into area (µm$^2$) using a calibration unit (0.8333). Mean of histone area was from the five independent fields (Supplementary Fig. 24) were denoted as NET histone area (µm$^2$).

**Preparation of macrophages**. For human macrophage preparation, peripheral blood mononuclear cells (PBMCs) were isolated from the whole blood of healthy human donors by standard density-gradient centrifugation with Ficoll-Paque (Amersham Biosciences). After centrifugation, the buffy coat containing leukocytes (PBMC) and platelets was further washed with PBS, and CD14$^+$ cells were purified using the VarioMACS technique with anti-CD14 microbeads (Miltenyi Biotec GmbH). Cells were then cultured in complete RPMI 1640 medium supplemented

with 10 ng/ml human M-CSF (R&D Systems) for 6 days[4]. For preparation of murine bone marrow-derived macrophages, bone marrow cells were isolated from femurs and tibias and cultured in RPMI 1640 complete medium supplemented with 10% (v/v) fetal calf serum (FCS) and 10 ng/ml of recombinant mouse M-CSF (R&D Systems) for 6–8 days. Human monocytes were obtained from healthy donors at the Taipei Blood Center of the Taiwan Blood Services Foundation, under a protocol (AS-IRB02-103202) approved by the IRB of the Clinical Center of the Department of Health, Taiwan. Written informed consent was obtained from all donors.

**Caspase activity**. The caspase-1 activity was measured after 6 h post bacterial stimulation, with a Caspase-1 Colorimetric Assay kit (BioVision, K111), whereas caspase-1 processing was detected by mouse caspase 1 ELISA kit to measure the amount of processed 20-kilodalton (p20) and 10-kilodalton (p10) caspase-1 in the culture supernatants (AdipoGen; AG-45B-0002-KI01).

**CLEC5A agonistic Ab synergizes with TLR ligands on NETosis**. Human neutrophils seeded on the coverslip were pre-incubated with agonistic CLEC5A mAb (clone: 2D8G9; 2.5 µg for $1 \times 10^6$ neutrophils) (Supplementary Fig. 25) for 10 min at room temperature, followed by cross-linking with a secondary antibody to mouse IgG (Jackson ImmunoResearch, 115-005-146) for 20 min, and then subjected to addition of TLR ligands at the amount of 30 µg per reaction for another 2 h NET components were detected by immunofluorescence staining using anti-MPO, anti-citrullinated histone H3 (cit-H3) antibodies and Hoechst 33342, followed by observation with a confocal microscope (Leica).

**Measurement of ROS production and NBT reduction assay**. Measurement of ROS was modified as described previously[49]. Briefly, human or mouse neutrophils ($1 \times 10^5$ cells per well) were suspend in HBSS plus Ca$_2^+$ and Mg$_2^{++}$, and incubated with live bacteria (m.o.i. = 3) for 30 min, followed by addition of pre-warmed (37 °C) mixture containing luminol (50 µM) and HRP (2 U per well) in HBSS to immediately prior to quantification. Luminescence was measured at an interval of 10 min till 120 min incubation at 37 °C in a luminometer (Titertek–Berthold). For the nitro blue tetrazolium (NBT) detection, neutrophils were seeded on the coverslip ($1 \times 10^5$ cells per well) for 1 h, followed by incubating the live bacteria (m.o.i. = 3) and NBT solution to a final concentration of 4 mM simultaneously. Cells were observed every 10 min till the appearance of indigo color (about 60-min post infection) and were fixed with 4% paraformaldehyde for 2 h at 4 °C. Cells were observed under a light microscopy and numbers of NBT positive were calculated for the average of five individual fields in each group.

**Measurement of bacterial killing and phagocytosis**. For phagocytosis analysis, mouse neutrophils ($1 \times 10^5$ cells per well) were seeded and infected with bacteria (m.o.i. = 0.1) for 60 min; supernatants were collected to determine the bacteria counts as the un-engulfed bacterial number. Percentage of phagocytosis was calculated as following formula: (1-unengulfed bacterial numbers/original bacterial numbers) × 100% engulfed bacterial numbers/original bacterial numbers) × 100%. For the killing assay, the formula is (1-engulfed bacterial numbers/original bacterial numbers- un-engulfed bacterial numbers) × 100%.

**Immunoblot analysis**. Mouse neutrophils ($3 \times 10^5$) incubated with live *L. monocytogenes* for 1 h were collected and lysed with lysis buffer (50 mM Tris pH 6.6, 10% SDS, 10% 2-Mercaptoethanol, 50% glycerol and 0.05% bromophenol blue). Immunoblot was firstly performed with anti-citH3 Ab (citrulline R2 + R8 + R17, ab51031, 1:1000), and then the used PVDF member were stripped and re-blotted with anti-H3 Ab (Sigma H0164, 1:1000) as an internal control.

**Monitoring bacterial growth in vivo**. A Xenogen IVIS Imaging System 200 series (PerkinElemer Inc.) was used to quantify luminescent live *L. monocytogenes* (Xen32) over time. Live lag-phase *L. monocytogenes* (Xen32) was washed and suspended as an amount of $5 \times 10^8$ CFUs in 100 µl saline, then i.v. injected into mice. To assess the effect of NETs on the *L. monocytogenes* dissemination, mice received an intraperitoneal injection of DNase I (4 KU) 1h before i.v. inoculation of Xen32. Bacterial dissemination was quantified by collectingblood at 2 and 4 h after bacterial inoculation. Mice were anesthetized with isofluorane while in the imaging chamber and imaged at 30 min, 1, 2 and 4 h after bacterial inoculation. Photons were measured during 60-s exposure, and total photon emissions from uniform area of each mouse were quantified by using the Living Image software (Caliper Life Science). At 4 h, the mice were killedand liver and spleen were homogenized and plated on TSA agar plates containing kanamycin and grown overnight to quantify bacterial CFUs.

**Experimental listeriosis**. Systemic Listeriosis model was performed with *L. monocytogenes* 10403S strain in 10- to 12-weeks-old mice. All groups of mice were i.v. injected with $1 \times 10^5$ CFUs of *L. monocytogenes* and monitored for 15 days. For the DNase I treatment, mice were intraperitoneally with DNase I (2 KU) every 2 days. Blood and liver tissue was collected as indicated time points for bacterial load determination. Livers were also collected and lysed by TRIzol to

extract total RNA for reverse-transcription into complementary DNA (cDNA) for analyzing gene expression. The primer sequences for real-time PCR analysis were listed in the Supplementary Table 3.

**Detection of leukocyte subpopulations in liver by FACS.** Mice were perfused with PBS buffer containing heparin (10 U/ml) prior tokilling. The livers were collected and minced into pieces followed by being digested with the buffer containing 1 mg/ml collagenase type IV (Sigma C5138) and 300 U/ml DNAase I (Sigma D5025) at 37 °C for 30 min. Liver homogenates were passed through cell strainer (40 μm), and lysed with RBC lysis buffer to remove the remaining red blood cells. The homogenates were further resuspended in 40% (w/v) Percoll-HBSS solution and overlaid onto 70% (w/v) Percoll-HBSS, followed by centrifuging at 2000 rpm for 30 min. After centrifugation, hepatic mononuclear cells (MNC) were collected from the layer of interphase and washed with PBS to remove residual Percoll. The isolated hepatic MNC were resuspended in FACS buffer for the following staining. To determine the IL-17A producing T cells, MNC ($5 \times 10^5$/reaction) were incubated with a cocktail of surface marker antibodies including CD3-PE (clone 17A2), CD4-APC (clone GK1.5) and γδ TCR-Brilliant Violet 421(clone GL3). The stained cells were then were fixed and permeabilized with CytoFix/perm solution (BD Bioscience 554714), followed by washing with BD wash buffer and staining with anti-IL17A-PE.Cy7 (clone TC11-18H10) and IFN-γ-Alexa Flour 700 (clone XMG 1.2) for 30 min on ice in the dark. After washing with FACS buffer, cells were resuspended in 1×PBS and analysed with BD FACSVerse. To determine monocytes and neutrophils, MNC were incubated with a cocktail of surface markers, including CD11b-Alexa Fluor 647 (clone M1/70), Ly6G-FITC (clone 1A8), Ly6C-PE (clone AL-21), CCR2-Alexa Flour 700 (clone SA203G11) and CX3CR1-PerCP.Cy5.5 (clone SA011F11). The stained cells were then were fixed and permeabilized with CytoFix/perm solution, followed by washing with BD wash buffer and staining with anti- IL-1β-PerCP (clone B122), TNF-PE (clone MP6-XT22) and iNOS-FITC (clone 6/iNOS). Stained cells washed with FACS buffer and resuspend in the PBS for analysing the subpopulation with BD FACSVerse. All the data were processed using FlowJo software (version 10, Treestar).

**Immunohistochemistry.** Formalin-fixed paraffin embedded liver sections were deparaffinized in xylene, rehydrated in graded alcohols and washed in distilled water. Antigen retrieval for neutrophil binding epitopes was performed by boiling sections in citrate buffer (10 mM, pH 6.0) for 24 min, follow by incubating the slides with trypsin (1 mg/ml) at 37 °C for 15 min. To block endogenous peroxidase activity, slides were incubated with 3% $H_2O_2$.methanol. Liver sections were washed and incubated with 10% BSA in PBS containing biotin (25% v/v) and avidin (25% v/v) to minimize non-specific binding of primary antibody. Slides were then incubated with primary antibody, anti-Ly6B.2 (1:200, clone: 7/4, Biorad) or isotype control (RIgG2a, 1:200) overnight at 4 °C. After washing, the sections were incubated with diluted secondary antibody (rabbit anti-rat IgG, 1:200, Vector BA4001), followed by VECTASTAIN ABC reagent (Vector PK6100). Slides were developed with chromogen diaminobenzidine (DAB, DAKO K3468) and counterstained with Gills Haematoxylin. Stained sections were digitized using Aperio Digital Pathology System and analyzed with ImageScope V9 softeware.

**Liver histological scores and assessment of liver injury.** The liver lobes were collected from experimental animals and placed in neutral buffered formalin (Sigma-Aldrich). The formalin-fixed tissue was then embedded in paraffin, cut in 5-microm sections, and stained with hematoxylin and eosin. For the degree of liver, inflammation was determined by a blinded histopathology score as previous study[50]. In brief, a score of 1 indicates that the number of micro abscesses on each liver section was below 10 and no necrosis region was found. A score of 2 indicates that the number of micro-abscesses on each liver section was >10 but <20 and no necrosis region was found. A score of 3 indicates that the number of micro-abscesses on each liver section was >20 but <30 and no necrosis region was found. A score of 3 indicates that the number of micro-abscesses on each liver section was >20 but <30 and no necrosis region was found. A score of 4 indicates that the number of micro-abscesses on each liver section was over 30 and no necrosis region was found. A score of 5 indicates that the number of necrosis region was below 5. A score of 6 indicates that the number of necrosis region was >5 but <10. A score of 7 indicates that the number of necrosis region was >10 but <15. A score of 8 indicates that the number of necrosis region was >15. The average score in each group was generated by the examination of liver sections from 5 mice. Serum levels of alanine ALT were detected in blood obtained via cheek pouch bleeding and as an indicator of hepatocellular injury. Samples were analyzed with Fuji Dri-Chem 4000i and results are expressed as international units per liter of serum.

**Microbial glycan microarray screening.** The interaction of CLEC5A.Fc with the microbial glycan array was determined in Dr. Richard Cumming's laboratory, as described previously[42]. The Fc-tagged CLEC5A sample and IgG control were diluted to 200 mg/ml in TSM binding buffer (20 mM Tris-HCl pH 7.4, 150 mM NaCl, 2 mM $CaCl_2$, 2 mM $MgCl_2$, 0.05% v/v Tween-20, and 1% w/v BSA), and 100 ml was added to the surface of the Microbial glycan microarray (MGM) slide. After placing a coverslip over the surface, samples were incubated for 1 h at room temperature in the dark. Slides were washed 4× in TSM wash buffer 1 (20 mM Tris-HCl pH 7.4, 150 mM NaCl, 2 mM $CaCl_2$, 2 mM $MgCl_2$, 0.05% v/v Tween-20)

and 4× in TSM wash buffer 2 (20 mM Tris-HCl pH 7.4, 150 mM NaCl, 2 mM $CaCl_2$, 2 mM $MgCl_2$). Detection of binding was achieved with Alexa Fluor-488-labeled anti-human IgG (Invitrogen) diluted to 5 mg/ml in TSM binding buffer on the slide for 1 h at room temperature in the dark. Slides were washed 4× in TSM wash buffer 1, 4× in TSM wash buffer 2, and 4× in water. Binding was quantitated with a microarray scanner (Scan Array Express, PerkinElmer Lifer Sciences) and processed in terms of average relative fluorescence intensity (RFU) of 4 replicates using ImaGene software (BioDiscovery).

**Extraction of cell wall components from _L. monocytogenes_.** The preparation of cell wall and sugar component of _L. monocytogenes_ was as described previously[51, 52]. Briefly, _L. monocytogenes_ (0.5 g) was ground and subjected to alkaline lysis (0.5 N NaOH, 2.0 ml, 60 °C, 8 h), followed by neutralization with 1 M AcOH (1.0 ml) and removal of solvent by a rotavapor. The precipitate was resuspended in MeOH (5 ml × 2), followed by centrifugation at room temperature for 30 min (4000 rpm) to collectsupernatant (MeOH-soluble fraction). The undissolved precipitate was resuspended with $H_2O$ (5 ml × 2), followed by centrifugation at room temperature for 30 min (4000 rpm) to collect the supernatant, followed by dialysis in distilled water ($H_2O$-soluble fraction). The collected fractions were lyophilized and used for TA sugar compositional analysis.

**Binding assay.** To determine CLEC5A binding to cell wall components, bacteria cell wall extracts (100 μg/ml fraction; 100 μl per well) were incubated on Maxi-Sorp ELISA plate (Nunc #469949) overnight at 4 °C, followed by addition of CLEC5A.Fc fusion protein (0–200 pM). The bound CLEC5A.Fc fusion protein was detected by HRP conjugated anti-human IgG antibody with 3,3′,5,5′-tetramethylbenzidine (TMB) (BD Pharmingen) substrate. In this ELISA assay, fusion protein and secondary antibody were diluted in a commercial protein-free blocking buffer (Thermo #37585). For cell- based binding assay, bacteria were labeled with Syto 60® (molecular probe S11342) at the finial concentration of 5 μM for 30 min at room temperature, followed by incubation with 293T cells ($1 \times 10^5$) overexpressing human CLEC5A and TLR2, respectively, at 4 °C for 30 min. Samples were washed with PBS to remove unbound bacteria, followed by flow cytometry analysis to detect bacteria-cell interaction.

**Statistical analysis.** All the data were presented as mean ± SEM and analyzed using GraphPad Prism software (Version 5.0). An unpaired, two-tailed student's $t$-test (for parametric data) or Mann–Whittney test (for nonparametric data) was used to determine the significance between two sets of data. When more than two groups were compared, a one-way ANOVA with the post hoc Bonferroni test (for parametric data) or a Kruskai–Wallis with post hoc Dunn's test (for nonparametric data) was used for multiple comparisons. Kaplan–Meier survival curves with log rank were used for analyzing mouse survival.

**Data availability.** All the data generated or analyzed during this study are available within the article and Supplementary Information files, or available from the authors upon request.

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

# ARTICLE

12. Pamer, E. G. Immune responses to Listeria monocytogenes. *Nat. Rev. Immunol.* **4**, 812–823 (2004).
13. Brinkmann, V. et al. Neutrophil extracellular traps kill bacteria. *Science* **303**, 1532–1535 (2004).
14. Hamada, S. et al. IL-17A produced by gammadelta T cells plays a critical role in innate immunity against listeria monocytogenes infection in the liver. *J. Immunol.* **181**, 3456–3463 (2008).
15. Edelson, B. T. & Unanue, E. R. MyD88-dependent but Toll-like receptor 2-independent innate immunity to Listeria: no role for either in macrophage listericidal activity. *J. Immunol.* **169**, 3869–3875 (2002).
16. Witte, C. E. et al. Innate immune pathways triggered by Listeria monocytogenes and their role in the induction of cell-mediated immunity. *Adv. Immunol.* **113**, 135–156 (2012).
17. Sahoo, M., Ceballos-Olvera, I., del Barrio, L. & Re, F. Role of the inflammasome, IL-1beta, and IL-18 in bacterial infections. *Sci. World J.* **11**, 2037–2050 (2011).
18. Mills, K. H. & Dunne, A. Immune modulation: IL-1, master mediator or initiator of inflammation. *Nat. Med.* **15**, 1363–1364 (2009).
19. Davis, B. K., Wen, H. & Ting, J. P. The inflammasome NLRs in immunity, inflammation, and associated diseases. *Annu. Rev. Immunol.* **29**, 707–735 (2011).
20. Gross, O. et al. Syk kinase signalling couples to the Nlrp3 inflammasome for anti-fungal host defence. *Nature* **459**, 433–436 (2009).
21. Rathinam, V. A. et al. The AIM2 inflammasome is essential for host defense against cytosolic bacteria and DNA viruses. *Nat. Immunol.* **11**, 395–402 (2010).
22. Wu, J., Fernandes-Alnemri, T. & Alnemri, E. S. Involvement of the AIM2, NLRC4, and NLRP3 inflammasomes in caspase-1 activation by Listeria monocytogenes. *J. Clin. Immunol.* **30**, 693–702 (2010).
23. Kim, S. et al. Listeria monocytogenes is sensed by the NLRP3 and AIM2 inflammasome. *Eur. J. Immunol.* **40**, 1545–1551 (2010).
24. Parker, H., Dragunow, M., Hampton, M. B., Kettle, A. J. & Winterbourn, C. C. Requirements for NADPH oxidase and myeloperoxidase in neutrophil extracellular trap formation differ depending on the stimulus. *J. Leukoc. Biol.* **92**, 841–849 (2012).
25. Flo, T. H. et al. Human toll-like receptor 2 mediates monocyte activation by Listeria monocytogenes, but not by group B streptococci or lipopolysaccharide. *J. Immunol.* **164**, 2064–2069 (2000).
27. Wang, Y. et al. Human PAD4 regulates histone arginine methylation levels via demethylimination. *Science* **306**, 279–283 (2004).
28. Li, P. et al. PAD4 is essential for antibacterial innate immunity mediated by neutrophil extracellular traps. *J. Exp. Med.* **207**, 1853–1862 (2010).
29. Gregory, S. H. & Wing, E. J. Neutrophil-Kupffer cell interaction: a critical component of host defenses to systemic bacterial infections. *J. Leukoc. Biol.* **72**, 239–248 (2002).
30. Conlan, J. W. Early host-pathogen interactions in the liver and spleen during systemic murine listeriosis: an overview. *Immunobiology.* **201**, 178–187 (1999).
31. Gregory, S. H., Sagnimeni, A. J. & Wing, E. J. Bacteria in the bloodstream are trapped in the liver and killed by immigrating neutrophils. *J. Immunol.* **157**, 2514–2520 (1996).
32. Conlan, J. W. & North, R. J. Neutrophil-mediated dissolution of infected host cells as a defense strategy against a facultative intracellular bacterium. *J. Exp. Med.* **174**, 741–744 (1991).
33. Sutton, C. E. et al. Interleukin-1 and IL-23 induce innate IL-17 production from gammadelta T cells, amplifying Th17 responses and autoimmunity. *Immunity* **31**, 331–341 (2009).
34. Serbina, N. V. & Pamer, E. G. Monocyte emigration from bone marrow during bacterial infection requires signals mediated by chemokine receptor CCR2. *Nat. Immunol.* **7**, 311–317 (2006).
35. Papayannopoulos, V., Metzler, K. D., Hakkim, A. & Zychlinsky, A. Neutrophil elastase and myeloperoxidase regulate the formation of neutrophil extracellular traps. *J. Cell Biol.* **191**, 677–691 (2010).
36. Kawakami, T. et al. Rab27a is essential for the formation of neutrophil extracellular traps (NETs) in neutrophil-like differentiated HL60 cells. *PLoS ONE* **9**, e84704 (2014).
37. Neeli, I., Khan, S. N. & Radic, M. Histone deimination as a response to inflammatory stimuli in neutrophils. *J. Immunol.* **180**, 1895–1902 (2008).
38. Gupta, A. K., Giaglis, S., Hasler, P. & Hahn, S. Efficient neutrophil extracellular trap induction requires mobilization of both intracellular and extracellular calcium pools and is modulated by cyclosporine A. *PLoS ONE* **9**, e97088 (2014).
39. Ali, A. et al. IL-1 Receptor Antagonist Treatment Aggravates Staphylococcal Septic Arthritis and Sepsis in Mice. *PLoS ONE* **10**, e0131645 (2015).
40. Galloway, J. B. et al. The risk of serious infections in patients receiving anakinra for rheumatoid arthritis: results from the British society for rheumatology biologics register. *Rheumatology* **50**, 1341–1342 (2011).
41. Miller, L. S. et al. MyD88 mediates neutrophil recruitment initiated by IL-1R but not TLR2 activation in immunity against Staphylococcus aureus. *Immunity* **24**, 79–91 (2006).
42. Stowell, S. R. et al. Microbial glycan microarrays define key features of host-microbial interactions. *Nat. Chem. Biol.* **10**, 470–476 (2014).
43. Aoki, N., Zganiacz, A., Margetts, P. & Xing, Z. Differential regulation of DAP12 and molecules associated with DAP12 during host responses to mycobacterial infection. *Infect. Immun.* **72**, 2477–2483 (2004).
44. Lockhart, E., Green, A. M. & Flynn, J. L. IL-17 production is dominated by gammadelta T cells rather than CD4 T cells during mycobacterium tuberculosis infection. *J. Immunol.* **177**, 4662–4669 (2006).
45. Eugster, M. R. & Loessner, M. J. The Listeria cell wall and associated carbohydrate polymers. *Methods. Mol. Biol.* **1157**, 129–140 (2014).
46. Lei, X. H., Fiedler, F., Lan, Z. & Kathariou, S. A novel serotype-specific gene cassette (gltA-gltB) is required for expression of teichoic acid-associated surface antigens in Listeria monocytogenes of serotype 4b. *J. Bacteriol.* **183**, 1133–1139 (2001).
47. Promadej, N., Fiedler, F., Cossart, P., Dramsi, S. & Kathariou, S. Cell wall teichoic acid glycosylation in Listeria monocytogenes serotype 4b requires gtcA, a novel, serogroup-specific gene. *J. Bacteriol.* **181**, 418–425 (1999).
48. Ermert, D. et al. Mouse neutrophil extracellular traps in microbial infections. *J. Innate Immun.* **1**, 181–193 (2009).
49. Wellington, M., Dolan, K. & Krysan, D. J. Live Candida albicans suppresses production of reactive oxygen species in phagocytes. *Infect. Immun.* **77**, 405–413 (2009).
50. Lin, Y. T., Liu, C. J., Yeh, Y. C., Chen, T. J. & Fung, C. P. Ampicillin and amoxicillin use and the risk of Klebsiella pneumoniae liver abscess in Taiwan. *J. Infect. Dis.* **208**, 211–217 (2013).
51. Kamisango, K. et al. Structural and immunochemical studies of teichoic acid of Listeria monocytogenes. *J. Biochem.* **93**, 1401–1409 (1983).
52. Fiedler, F. Biochemistry of the cell surface of Listeria strains: a locating general view. *Infection* **16**(Suppl 2), S92–S97 (1988).

## Acknowledgements

We thank Drs John Kao, Caroline Milner, Fu-Tong Liu, Jen-Tsan Ashley Chi, and Judith Chou for critical comments. We are grateful for Drs Wei-Chieh, Cheng, and Tsui-Ling Hsu provides peptidoglycan and technical assistance. Dr Richard Cumming, Dr Chih-Ya Yang, Xing-Jei Lin, I-Shuen Tsai, Sung Pei-Shen, and Li-Wen Lo for technical assistance. We are indebted to Dr Daniel Portnoy, Dr Sean Stowell, Dr Martin Loessner, Dr Yang Shen, and Dr Fu-Tong Liu for sharing various serogroups of *L. monocytogenes*. We are also appreciative to the technical services provided by the "Transgenic Mouse Model Core Facility of the National Core Facility Program for Biotechnology, National Science Council" and technical services provided by Flow cytometry Core Facility of National Yang Ming University. CLEC5A-microbial glycan interaction was performed by the Consortium for Functional Glycomics (Richard D. Cummings). This work was supported by Academia Sinica and Ministry of Science and Technology (MOST 106-2321-B-001-037, MOST 103-2320-B-001-010-MY3, MOST 104-2320-B-010-0440-MY3, Taipei Medical University (TMU102-AE1-B26), Yen Tjing Ling Medical Foundation (CI-106-16) and Summit and Thematic Research Projects.

## Author contributions

S.-T.C. designed and performed experiments, data analysis, and wrote the paper; F.-J.L., T.-Y.C., T.-Y.H., and W.-Y.L., performed experiments; S.-M.L. provided mice, technical support. T.-Y.C., Y.-C.Y. pathological studies; M.H., P.-S.S. and W.-B.Y. for technical support, S.-L.H. was involved in the experiment designing, data analysis, and manuscript writing.

## Additional information

**Competing interests:** The authors declare no competing financial interests.

