## [Peer review file · Nature Communications]

Reviewers' comments:

Reviewer #1 (Remarks to the Author):

The authors identify a role for CLEC5A in control of systemic *L. monocytogenes* infection. CLEC5A knock out mice exhibited increased mortality due to increased bacterial loads in the blood and liver and severe liver necrosis. In addition, the authors observed increased infiltration of CCR2+ inflammatory monocytes, a reduction of IL-17A-secreting $\gamma\delta$ T cells and decreased production of pro-inflammatory cytokines, IL-1 β and IL17A in the liver of CLEC5A deficient mice. Furthermore CLEC5A deficient neutrophils exhibited reduced ROS production and NET formation in response to *L. monocytogenes*. The authors also identify co-localisation of CLEC5A and TLR2 and propose that co-activation of these receptors enhances the immunity against bacterial infection.

Overall, this study suggests that CLEC5A plays a pivotal role in the innate immune response to bacterial infection, but the authors need to address the following issues:

- 1) Does Clec5a actually recognise *Listeria*? FACS with fusion proteins or reporter cells should answer this question. Is the requirement of TLR2 direct or indirect (eg: is it required for induction of CLEC5a upon infection...that may explain the partial phenotype observed in these mice) ? Do DAP12 cells/mice phenocopy the CLEC5a ko mice?
- 2) Which cell population is primarily responsible for the phenotype? The data show that the absence of NET formation is critical, suggesting PMN function is the primary protective cell type. Is adoptive transfer of WT PMN into KO protective? What does the inflammatory infiltrate look like the DNase treated wt mice? Are these monocytes? Are the functions of these monocytes (phagocytosis, killing) impaired in the absence of CLEC5a? Can the authors demonstrate loss of NET formation in vivo?
- 3) Do gamma-delta T-cells express CLEC5a (like they do Dectin-1) and could this explain the lack of IL-17 producing cells?
- 4) Is the expression of CLEC5A similar on mouse and human cells?
- 5) Is the effect on inflammasome activation and Il-1beta production specific to *Listeria*. Controls are needed here...see also point 8 below.
- 6) Supplementary Figure 4 – NET formation in the liver of *L. monocytogenes* infected mice. It is not clear that TLR2 KO mice have reduced Cit-H3 as the authors claim. CLEC5A KO mice have some reduction in Cit-H3. DNase treatment of WT mice does not make Cit-H3 undetectable (line 178). These types of data should be quantified throughout the manuscript.
- 7) Figure 4C – Level of significance of TNF α downregulation in WT/DNase1, CLEC5A and TLR2 KO mice is not shown on the graph.
- 8) The introduction of new data in the discussion is not appropriate. The effect on *S. aureus* suggests that CLEC5a might have a much broader role in antibacterial immunity. Is there a role for this receptor in control of gram negative infections?
- 9) In figure 6, are agonistic antibodies against other similar activation CLR's able to trigger similar responses, and can they be used to rescue the phenotype in vivo?
- 10) There are many typographical errors throughout.

Reviewer #2 (Remarks to the Author):

Listeria monocytogenes, the causative agent of listeriosis, is a model pathogen for studying the mechanisms of the immune response in vertebrates. Both, the innate as well as the adaptive immune response play an important role in the recognition and elimination of this pathogen. This study examined the role of CLEC5A, a pattern recognition receptor (PRR), during *L. monocytogenes* infection.

The authors provide evidence that CLEC5A is crucial for detection and elimination of *L. monocytogenes*, a function which is more important than the previously described function of TLR2. In addition, they show that the co-activation of both PRRs, CLEC5A and TLR2, enhances host

immunity against *L. monocytogenes*.

While this is an important study on the role of CLEC5A in infection, the study would have been strengthened by the detection of the glucan moiety recognized by this receptor.

Comments:

- CLEC5A regulation was previously described for viral infection as well as for mycobacterial infection. The authors show that CLEC5A not only plays a role during infection with both *L. monocytogenes* and *Staphylococcus aureus* suggesting that it is a general player in the recognition and elimination of bacteria. Thus, the authors should mention that this PRR was already previously associated with bacterial infection (*Mycobacterium bovis*; Aoki et al., 2004).
- The authors should provide more detail on how they have performed the CLEC5A-microbial glycan interaction assay (Supplementary Table 2). In addition, they should perform this assay with *L. monocytogenes* to detect and characterize the glycan(s) that interact(s) with CLEC5A.
- Following detection of the CLEC5A-*L. monocytogenes* glycan, the authors should infect cells as well as mice with a *L. monocytogenes* non-producing glycan mutant and show the abolished activation of CLEC5A-dependent anti-Listerial immune response (ROS production, NET formation, cytokine expression and bacterial numbers in the liver).
- The authors should check the spelling in the manuscript and pay attention to the nomenclature used for the mouse lineages employed.
- Figure 3A: Why are there no fluorescent bacteria observed in the tail vein of *Clec5a*^{-/-} infected mice like the others?

Reviewer #1 (Remarks to the Author):

Overall, this study suggests that CLEC5A plays a pivotal role in the innate immune response to bacterial infection, but the authors need to address the following issues:

- 1) Does Clec5a actually recognize Listeria? a) FACS with fusion proteins or reporter cells should answer this question. b) Is the requirement of TLR2 direct or indirect (eg: is it required for induction of CLEC5a upon infection...that may explain the partial phenotype observed in these mice) ? c) Do DAP12 cells/mice phenocopy the CLEC5a ko mice?**

Answers:

- a) *We showed that CLEC5A.Fc binds to L. monocytogenes cell walls components directly (Supplementary figure 23 a), and fluorochrome-labeled L. monocytogenes binds to 293T cells transfected with CLEC5A directly (Supplementary figure 23 b).*
- b) *To address this question, WT and TLR2^{-/-} leukocytes were incubated with alive Listeria monocytogenes (MOI= 10) for 1 hour, 2 hour, and 3 hours, respectively, followed by examining CLEC5A expression by flow cytometry (please see the representative figure in the following). We found that CLEC5A was upregulated in both WT and TLR2 knockout leukocytes (CD11b⁺Ly6c⁺ and CD11b⁺Ly6g⁺ cells), though the basal level of CLEC5A was slightly higher in TLR2^{-/-} leukocytes. This observation suggests that upregulation of CLEC5A by L. monocytogenes is TLR2-independent, and the higher CLEC5A upregulation level in TLR2^{-/-} leukocytes may partially rescue the functional impairment of TLR2^{-/-} leukocytes*
- c) *The DAP12-deficient macrophages/neutrophils have similar phenotype as CLEC5A^{-/-} macrophages (Supplementary figure 1), and DAP12^{-/-} mice are as susceptible as CLEC5A^{-/-} mice after L. monocytogenes infection (Supplementary figure 19).*

Figure: Upregulation of CLEC5A after Listerial infection is TLR2-independent. (a) Wild type and (b) TLR2^{-/-} mouse whole blood cells (3×10^6) were incubated with or without *L. monocytogenes* (3×10^7) for 2 h, followed by addition of lysis buffer to disrupt red blood cells before incubation with fluorochrome-conjugated antibodies to CD11b, Ly6g, Ly6c, and CLEC5A. The mean fluorescence intensity (MFI) of CLEC5A was determined by flow cytometry.

d) *The DAPI2-deficient macrophages/neutrophils has similar phenotype as CLEC5A^{-/-} macrophages (Supplementary figure 1), and DAPI2^{-/-} mice is as susceptible as CLEC5A^{-/-} mice after L. monocytogenes infection (Supplementary figure 19).*

- 2) Which cell population is primarily responsible for the phenotype? The data show that the absence of NET formation is critical, suggesting PMN function is the primary protective cell type. a) Is adoptive transfer of WT PMN into KO protective? b) What does the inflammatory infiltrate look like in the DNase-treated WT mice? Are these monocytes? c) Are the functions of these monocytes (phagocytosis, killing) impaired in the absence of CLEC5a? d) Can the authors demonstrate loss of NET formation in vivo?

Answers:

- a) *YES. Adoptive transfer of WT PMN into CLEC5A^{-/-} mice reduced L. monocytogenes loads in blood and liver (Supplementary Fig. 16); and CLEC5A^{-/-} mice with adoptive transfer of WT PMN showed less weight loss and high vitality than those without transfer.*
 - b) *The inflammatory infiltrate of the DNase I-treated WT mice is similar to the CLEC5A^{-/-} mice post L. monocytogenes infection at day 5 (Supplementary Fig. 15). The major population of infiltrating cell is monocytes (Supplementary Fig. 15).*
 - c) *CLEC5A^{-/-} CCR2⁺ inflammatory monocytes expressed much lower inducible nitric oxidase synthase (iNOS) (Supplementary Fig. 13) in liver, indicating the ROS production and cytotoxic effect of this population is impaired in vivo.*
 - d) *As we showed previously, loss of NET formation was observed in the knockout mice, and NET is not detectable after DNase-I treatment in mice (Supplementary Fig. 5).*
- 3) Do gamma-delta T-cells express CLEC5a (like they do Dectin-1) and could this explain the lack of IL-17 producing cells?**

Answers:

CLEC5A is not detectable in IL17-producing gamma/delta T cells (Supplementary Fig. 8)

4) Is the expression of CLEC5A similar on mouse and human cells?

Answers:

Yes. Both human and mice CLEC5A are expressed in neutrophils, monocytes, and macrophages abundantly (Supplementary Fig. 8).

5) Is the effect on inflammasome activation and IL-1beta production specific to Listeria.

Controls are needed here...see also point 8 below.

Answers:

We showed that CLEC5A is also involved in the inflammasome activation and IL-1 β production induced by S. aureus (Supplementary Fig. 20). It may be also involved in other bacteria-induced inflammasome activation based on its interaction with other bacterial extracts using pathogen glycan array (Supplementary Fig. 23, Supplementary table I).

- 6) Supplementary Figure 4 – NET formation in the liver of L. monocytogenes infected mice. It is not clear that TLR2 KO mice have reduced Cit-H3 as the authors claim. CLEC5A KO mice have some reduction in Cit-H3. DNase treatment of WT mice does not make Cit-H3 undetectable (line 178). These types of data should be quantified throughout the manuscript.**

Answers:

- a) *Although the average NET Cit-H3 area in TLR2 KO mice (16580 μm^2) was a little bit less than that of in WT mice liver (23972 μm^2), there is no significant difference statistically. (Supplementary Fig. 5b).*
- b) *Quantitation was performed as shown in (Supplementary Fig. 5b).*

7) Figure 4C – Level of significance of TNF α downregulation in WT/DNAse1, CLEC5A and TLR2 KO mice is not shown on the graph.

Answers:

*We already added the “***” on the chart (Figure 4c, most left panel). We are grateful for your kindness to remind us for this mistake.*

8) a) The introduction of new data in the discussion is not appropriate. b) The effect on *S. aureus* suggests that CLEC5a might have a much broader role in antibacterial immunity. c) Is there a role for this receptor in control of gram negative infections?

Answers:

- a) *We followed the suggestions and moved the results regarding the issue of “Dissection of CLEC5A/TLR2 co-activated signaling pathways” and “Effect of CCR2⁺ inflammatory monocytes in CLEC5A^{-/-} mice” from the DISCUSSION session to RESULT session (page 12 and page 13, supplementary figures 10-13, highlighted in yellow).*
- b) *We added a paragraph and new data regarding the role of CLEC5A in host defense against other Gram-positive and Gram-negative bacteria at the DISCUSSION session (Page 17, Supplementary Fig. 20 & 21, highlighted in yellow).*
- c) *Results from the experiment suggest that CLEC5A plays a significant role to control Gram-positive and Gram-negative bacteria infection (Supplementary Fig. 20 & 21).*

9) In figure 6, are agonistic antibodies against other similar activation CLR's able to trigger similar responses, and can they be used to rescue the phenotype in vivo?

Answers:

*Activation of other similar myeloid CLR's by curdlan (Dectin-1 ligand) or UV-inactivated *Candida* (dectin-1 and dectin-2 ligands) were unable to rescue the functional impairment of CLEC5A^{-/-} leukocytes against *L. monocytogenes*. This observation suggests that the downstream signaling of CLEC5A (DAP12) is different from Dectin-1 (intrinsic ITAM motif in cytoplasmic region) and Dectin-2 (FcR γ -coupled receptor).*

10) There are many typographical errors throughout.

Answer:

We check the spelling carefully and correct the typographical errors.

Reviewer #2 (Remarks to the Author):

Comments:

- 1) **CLEC5A regulation was previously described for viral infection as well as for mycobacterial infection. The authors show that CLEC5A not only plays a role during infection with both *L. monocytogenes* and *Staphylococcus aureus* suggesting that it is a general player in the recognition and elimination of bacteria. Thus, the authors should mention that this PRR was already previously associated with bacterial infection (*Mycobacterium bovis*; Aoki et al., 2004).**

Answer:

We mentioned this discovery and cite this reference in the DISCUSSION section of this revised manuscript (Line 2-5, page 18)

- 2) **a) The authors should provide more detail on how they have performed the CLEC5A microbial glycan interaction assay (Supplementary Table 2). b) In addition, they should perform this assay with *L. monocytogenes* to detect and characterize the glycan(s) that interact(s) with CLEC5A.**

Answer:

a) The original Supplementary Table 2 is changed as Supplementary Table 1 in the revised manuscript, and more detail description of CLEC5A.Fc binding to pathogen glycan array is included (Materials and Methods; Line 13-26, page 29).

*b) We also purify cell wall components and glycans from *L. monocytogenes* and performed binding assay (Supplementary figure 23), and the description is inserted in the DISCUSSION session (Line 6-27, page 18), and the method is shown in the 'Materials and Methods' (page 30).*

- 3) **Following detection of the CLEC5A-*L. monocytogenes* glycan, the authors should infect cells as well as mice with a *L. monocytogenes* non-producing glycan mutant and show the abolished activation of CLEC5A-dependent anti-Listerial immune response (ROS production, NET formation, cytokine expression and bacterial numbers in the liver).**

Answer:

*a) We consulted expert in *L. monocytogenes* Professor Daniel A. Portnoy – (UC Berkeley) and check the published literature for the *L. monocytogenes* non-producing glycan mutant, and it seems to us that such kind of mutant is not available.*

- b) However, we do our best to get as many mutants and different serotypes with various glycans as possible (Supplementary Table 2). We found that these mutants have similar functions as WT (serotype1/2a) to induce NET formation and the production of cytokines and ROS. This information suggests that terminal glycans do not contribute significantly in CLEC5A recognition to cell wall.
- c) We also synthesize the N-acetylglucosamine (GlcNAc) and N-acetylmuramic acid (MurNAc) disaccharide, which comprise the backbone of *L. monocytogenes* backbone. We found that CLEC5A binding to *L. monocytogenes* cell walls were inhibited by peptidoglycan disaccharide (N-acetylglucosamine (GlcNAc) and N-acetylmuramic acid (MurNAc)), suggesting CLEC5A is mainly involved in recognition to peptidoglycan. This observation is supported by the fact that impaired of TNF production was observed in CLEC5A^{-/-} macrophages stimulated with peptidoglycan.
- d) We did notice there is L-form *L. monocytogenes*, which lack of whole cell wall. However, we are unable to obtain this bacterial strain, and L-form bacteria lack both glycan and all the other components on cell walls. Thus, we did not compare the effect of L form L-form bacteria with WT to activate neutrophils and macrophages.
- 4) The authors should check the spelling in the manuscript and pay attention to the nomenclature used for the mouse lineages employed.**

Answer:

We check the spelling of the nomenclature used for the mouse lineages carefully in this revised manuscript.

- 5) Figure 3A: Why are there no fluorescent bacteria observed in the tail vein of Clec5a^{-/-} infected mice like the others?**

Answer:

We repeated the experiments and the original figure was replaced with a new photo (Figure 3a). It did show the fluorescent bacteria in the tail vein of Clec5a^{-/-} mice. The previous one may be due to a) variation of bacteria amount for inoculation, or b) rapid spreading of bacteria after inoculation in Clec5a^{-/-} mice. After repeating the experiments several times, we can detect the fluorescent bacteria in the tail vein of Clec5a^{-/-} mice in most of the experiments.

We thank you in advance for your kind consideration.

REVIEWERS' COMMENTS:

Reviewer #1 (Remarks to the Author):

I am happy with the revised manuscript, and additional data added which greatly strengthens the story. Stylistically however, the discussion section still presents a great deal of results and I think these should be integrated into the results section.

There are still also numerous typographical error eg: producing in line 369, 371 and 372

Reviewer #2 (Remarks to the Author):

The authors have addressed all of the reviewers comments in an appropriate manner with the exception of using a glycan-negative *L. monocytogenes* mutant. However, the authors demonstrate that core components of peptidoglycan (GlcNAc and MurNAc) are recognized by CLEC5A.

The manuscript is recommended for publication.

中央研究院
基因體研究中心
Genomics Research Center
ACADEMIA SINICA

Genomics Research Center,
Academia Sinica
128, Academia Road, Section 2
Nankang, Taipei 115, Taiwan R.O.C.
Tel/+886-2-27871245
Fax/+886-2-2789-8811
E-mail: slhsieh@gate.sinica.edu.tw

REVIEWERS' COMMENTS:

Reviewer #1 (Remarks to the Author):

I am happy with the revised manuscript, and additional data added which greatly strengthens the story. Stylistically however, the discussion section still presents a great deal of results and I think these should be integrated into the results section.

There are still also numerous typographical error e.g: producing in line 369, 371 and 372

Answer:

1) We seriously consider his suggestion, but we do not move any paragraph from DISCUSSION SESSION into RESULT SESSION, because the data in DISCUSSION SESSION is used to support the findings in the RESULT SESSION, and insertion of these paragraphs would disrupt the fluency. Instead, we delete the 3rd paragraph of the DISCUSSION SESSION because it is redundant to what we describe in the RESULT SESSION.

2) The typo errors in line 369, 371 and 372 are located in the session we delete as mention above, thus not present in this revised version.

Reviewer #2 (Remarks to the Author):

The authors have addressed all of the reviewers comments in an appropriate manner with the exception of using a glycan-negative *L. monocytogenes* mutant. However, the authors demonstrate that core components of peptidoglycan (GlcNAc and MurNAc) are recognized by CLEC5A.

The manuscript is recommended for publication.

Answer:

Thank you very much for reviewer's comment.